# Linear $Q$-Learning Does Not Diverge in $L^2$: Convergence Rates to a Bounded Set

**Xinyu Liu** [* 1]  **Zixuan Xie** [* 1]  **Shangtong Zhang** [1]

## Abstract

$Q$-learning is one of the most fundamental reinforcement learning algorithms. It is widely believed that $Q$-learning with linear function approximation (i.e., linear $Q$-learning) suffers from possible divergence until the recent work Meyn (2024) which establishes the ultimate almost sure boundedness of the iterates of linear $Q$-learning. Building on this success, this paper further establishes the first $L^2$ convergence rate of linear $Q$-learning iterates (to a bounded set). Similar to Meyn (2024), we do not make any modification to the original linear $Q$-learning algorithm, do not make any Bellman completeness assumption, and do not make any near-optimality assumption on the behavior policy. All we need is an $\epsilon$-softmax behavior policy with an adaptive temperature. The key to our analysis is the general result of stochastic approximations under Markovian noise with fast-changing transition functions. As a side product, we also use this general result to establish the $L^2$ convergence rate of tabular $Q$-learning with an $\epsilon$-softmax behavior policy, for which we rely on a novel pseudo-contraction property of the weighted Bellman optimality operator.

## 1. Introduction

Reinforcement learning (RL, Sutton & Barto (2018)) emerges as a powerful paradigm for training agents to make decisions sequentially in complex environments. Among various RL algorithms, $Q$-learning (Watkins, 1989; Watkins & Dayan, 1992) stands out as one of the most celebrated (Mnih et al., 2015). The original $Q$-learning in Watkins

(1989) uses a look-up table for representing the action-value function. To improve generalization and work with large or even infinite state spaces, linear function approximation is used to approximate the action value function, yielding linear $Q$-learning.

Linear $Q$-learning is, however, widely believed to suffer from possible divergence (Baird, 1995; Sutton & Barto, 2018). In other words, the weights of linear $Q$-learning can possibly diverge to infinity as learning progresses. However, Meyn (2024) recently proves that when an $\epsilon$-softmax behavior policy with an adaptive temperature is used, linear $Q$-learning does not diverge. Instead, the weights are eventually bounded almost surely. Or more specifically, let $\{w_t\}$ be the weights of linear $Q$-learning. Meyn (2024) proves that $\limsup_t \|w_t\| < C$ almost surely for some deterministic constant $C$. Building on this success, we further provide a nonasymptotic analysis of linear $Q$-learning by **establishing the first $L^2$ convergence rate of linear $Q$-learning to a bounded set**. Specifically, we establish the rate at which $\mathbb{E}[\|w_t\|^2]$ diminishes to a bounded set. Notably, this work differs from many previous analyses of linear $Q$-learning (prior to Meyn (2024)) in that, except for the aforementioned behavior policy, we do not make any modification to the original linear $Q$-learning algorithm (e.g., no target network, no weight projection, no experience replay, no i.i.d. data, no regularization). We also use the weakest possible assumptions (e.g., no Bellman completeness assumption, no near-optimality assumption on the behavior policy). Table 1 summarizes the improvements.

Our $L^2$ convergence rate is made possible by a novel result concerning the **convergence of stochastic approximations under fast-changing time-inhomogeneous Markovian noise**, where the transition function of the Markovian noise evolves as fast as the stochastic approximation weights, i.e., there is only a single timescale. By contrast, Zhang et al. (2022) obtain similar results only with a two-timescale framework, where the transition function of the Markovian noise evolves much slower than the stochastic approximation weights.

As a side product, we also use this general stochastic approximation result to establish the $L^2$ convergence rate of tabular $Q$-learning with an $\epsilon$-softmax behavior policy. To

---
[*]Equal contribution [1]Department of Computer Science, University of Virginia. Correspondence to:
Xinyu Liu \<xinyuliu@virginia.edu\>,
Zixuan Xie \<xie.zixuan@email.virginia.edu\>,
Shangtong Zhang \<shangtong@virginia.edu\>.

*Proceedings of the 42$^{nd}$ International Conference on Machine Learning*, Vancouver, Canada. PMLR 267, 2025. Copyright 2025 by the author(s).

| | Type | Algorithm Modification | | | | Assumptions | | Behavior Policy | Rate |
|---|---|---|---|---|---|---|---|---|---|
| | | $\bar{w}_t$ | $\mathcal{B}$ | $\mathcal{H}$ | $\|w_t\|$ | $\mathcal{TX}\overline{w}$ | $\mathcal{X}$ | | |
| Melo et al. (2008) | linear | ✓ | ✓ | ✓ | ✓ | ✓ | ✓ | $\mu_*$ | |
| Lee & He (2020) | linear | ✓ | ✓ | ✓ | ✓ | ✓ | | $d_\mu$ | |
| Chen et al. (2022) | linear | ✓ | ✓ | ✓ | ✓ | ✓ | ✓ | $\mu_*$ | ✓ |
| Carvalho et al. (2020) | linear | | | ✓ | ✓ | ✓ | ✓ | $d_\mu$ | |
| Zhang et al. (2021) | linear | | ✓ | | | ✓ | ✓ | $\mu_{\epsilon,\text{softmax}}$ | |
| Gopalan & Thoppe (2022) | linear | ✓ | ✓ | ✓ | ✓ | ✓ | ✓ | $d_{\mu_{\epsilon,\text{arg max}}}$ | |
| Chen et al. (2023) | linear | | ✓ | | ✓ | ✓ | ✓ | $\mu$ | ✓ |
| Han-Dong & Donghwan (2024) | linear | ✓ | ✓ | ✓ | | ✓ | ✓ | $d_\mu$ | |
| Meyn (2024) | linear | ✓ | ✓ | ✓ | ✓ | ✓ | ✓ | $\mu_{\epsilon,\text{softmax}}$ | |
| Fan et al. (2020) | neural | | ✓ | | ✓ | | ✓ | $d_\mu$ | ✓ |
| Xu & Gu (2020) | neural | ✓ | ✓ | | ✓ | ✓ | ✓ | $\mu_*$ | ✓ |
| Cai et al. (2023) | neural | ✓ | ✓ | | ✓ | ✓ | ✓ | $d_{\mu_*}$ | ✓ |
| Zhang et al. (2023b) | neural | | | ✓ | ✓ | ✓ | ✓ | $\mu_{\epsilon,\text{arg max}}$ | ✓ |
| Theorem 1 | linear | ✓ | ✓ | ✓ | ✓ | ✓ | ✓ | $\mu_{\epsilon,\text{softmax}}$ | ✓ |

*Table 1.* Summary of notable analyses of linear $Q$-learning. We additionally include $Q$-learning with neural networks for completeness. The exact definitions of the terminologies used in the columns and a more detailed exposition of the table are in Section 4. "$\bar{w}_t$" is checked if target network is not used. "$\mathcal{B}$" is checked if experience replay is not used. "$\mathcal{H}$" is checked if weight projection is not used. "$\|w_t\|$" is checked if no additional regularization is used. "$\mathcal{TX}\overline{w}$" is checked if Bellman completeness assumption is not used. "$\mathcal{X}$" is checked if no restrictive assumptions on the features are used. "Rate" is checked if a convergence rate is provided. For the "Behavior Policy" column, "$\mu$" indicates that a fixed behavior policy is used. "$\mu_*$" indicates that a fixed behavior policy is used and some strong near-optimality assumption of the behaivor policy is made. "$\mu_{\epsilon,\text{arg max}}$" indicates that an $\epsilon$-greedy policy is used, "$\mu_{\epsilon,\text{softmax}}$" indicates that an $\epsilon$-softmax policy is used. Let $\mu_0 \in \{\mu, \mu_*, \mu_{\epsilon,\text{arg max}}, \mu_{\epsilon,\text{softmax}}\}$ be any type of the aforementioned behavior policy. Then $d_{\mu_0}$ indicates that i.i.d. samples from the stationary distribution of $\mu_0$ is provided directly instead of obtaining Markovian samples by executing $\mu_0$.

| | $\alpha_{\nu(S_t, A_t, t)}$ | Behavior Policy | Rate |
|---|---|---|---|
| Watkins (1989); Watkins & Dayan (1992); Jaakkola et al. (1993) Tsitsiklis (1994); Littman & Szepesvári (1996) | | any | |
| Szepesvári (1997) | | $d_\mu$ | ✓ |
| Even-Dar et al. (2003) | | any | ✓ |
| Lee & He (2020) | ✓ | $d_\mu$ | |
| Devraj & Meyn (2022) | ✓ | $\mu$ | |
| Chen et al. (2021); Li et al. (2024); Zhang & Xie (2024) | ✓ | $\mu$ | ✓ |
| Theorem 2 | ✓ | $\mu_{\epsilon,\text{softmax}}$ | ✓ |

*Table 2.* Summary of notable analyses of tabular $Q$-learning. The exact definitions of the terminologies used in the columns and a more detailed exposition of the table are in Section 4. "$\alpha_{\nu(S_t, A_t, t)}$" is checked if count based learning rate is not required. Notably, analyses of the synchronous variant of tabular $Q$-learning is not surveyed in this paper.

our knowledge, this is the first time that such convergence rate is established. The key to this success is the identification of **a novel pseudo-contraction property of the weighted Bellman optimality operator**. Table 2 summarizes the improvements over previous works.

## 2. Background

**Notations.** We use $\langle x, y \rangle \doteq x^\top y$ to denote the standard inner product in Euclidean spaces. A function $f$ is said to be $L$-smooth (w.r.t. some norm $\|\cdot\|_s$) if $\forall w, w'$,

$$f(w') \leq f(w) + \langle \nabla f(w), w' - w \rangle + \frac{L}{2} \|w' - w\|_s^2. \quad (1)$$

Since all norms in finite dimensional spaces are equivalent, when we say a function is smooth in this paper, it means it is smooth w.r.t. any norm (with $L$ depending on the choice of the norm). In particular, to simplify notations, we in this paper use $\|\cdot\|$ to denote an arbitrary vector norm such that its square $\|\cdot\|^2$ is smooth. We abuse $\|\cdot\|$ to also denote the corresponding induced matrix norm. We use $\|\cdot\|_*$ to denote the dual norm of $\|\cdot\|$. We use $\|\cdot\|_2$ and $\|\cdot\|_\infty$ to denote

the $\ell_2$ and infinity norm. We use functions and vectors exchangeably when it does not confuse. For example, $f$ can denote both a function $\mathcal{S} \to \mathbb{R}$ and a vector in $\mathbb{R}^{|\mathcal{S}|}$ simultaneously.

We consider an infinite horizon Markov Decision Process (MDP, Bellman (1957)) defined by a tuple $(\mathcal{S}, \mathcal{A}, p, r, \gamma, p_0)$, where $\mathcal{S}$ is a finite set of states, $\mathcal{A}$ is a finite set of actions, $p : \mathcal{S} \times \mathcal{S} \times \mathcal{A} \to [0, 1]$ is the transition probability function, $r : \mathcal{S} \times \mathcal{A} \to \mathbb{R}$ is the reward function, $\gamma \in [0, 1)$ is the discount factor, and $p_0 : \mathcal{S} \to [0, 1]$ denotes the initial distribution. At the time step 0, an initial state $S_0$ is sampled from $p_0$. At the time step $t$, an action $A_t$ is sampled according to some policy $\pi : \mathcal{A} \times \mathcal{S} \to [0, 1]$, i.e., $A_t \sim \pi(\cdot|S_t)$. A reward $R_{t+1} \doteq r(S_t, A_t)$ is then emitted and a successor state $S_{t+1}$ is sampled from $p(\cdot|S_t, A_t)$. We use $P_\pi$ to denote the transition matrix between state action pairs for an arbitrary policy $\pi$, i.e., $P_\pi[(s, a), (s', a')] = p(s'|s, a)\pi(a'|s')$. We use $q_\pi : \mathcal{S} \times \mathcal{A} \to \mathbb{R}$ to denote the action value function of a policy $\pi$, which is defined as $q_\pi(s, a) \doteq \mathbb{E}_\pi \left[ \sum_{i=0}^\infty \gamma^i R_{t+i+1} | S_t = s, A_t = a \right]$. One fundamental task in RL is control, where the goal is to find the optimal action value function, denoted as $q_*$, satisfying $q_*(s, a) \geq q_\pi(s, a) \forall \pi, s, a$. It is well-known that the $q_*$ is the unique fixed point of the Bellman optimality operator $\mathcal{T} : \mathbb{R}^{|\mathcal{S}||\mathcal{A}|} \to \mathbb{R}^{|\mathcal{S}||\mathcal{A}|}$ defined as $(\mathcal{T}q)(s, a) \doteq \sum_{s'} p(s'|s, a) \left[ r(s, a) + \gamma \max_{a'} q(s', a') \right]$.

**Tablular $Q$-Learning.** $Q$-learning is the most celebrated method to estimate $q_*$. In its simplest form, it uses a lookup table $q \in \mathbb{R}^{|\mathcal{S}||\mathcal{A}|}$ to store the estimate of $q_*$ and generates the iterates $\{q_t\}$ as

$$A_t \sim \mu_{q_t}(\cdot|S_t), \qquad \text{(tabular } Q\text{-learning)}$$
$$\delta_t = R_{t+1} + \gamma \max_{a'} q_t(S_{t+1}, a') - q_t(S_t, A_t),$$
$$q_{t+1}(S_t, A_t) = q_t(S_t, A_t) + \alpha_t \delta_t.$$

Here, $\{\alpha_t\}$ are learning rates and $\mu_q$ is the behavior policy. In this paper, we consider an $\epsilon$-softmax behavior policy defined as

$$\mu_q(a|s) \doteq \frac{\epsilon}{|\mathcal{A}|} + (1 - \epsilon)\frac{\exp(q(s,a))}{\sum_b \exp(q(s,b))}, \qquad (2)$$

where $\epsilon \in (0, 1]$ controls the degree of exploration.

**Linear $Q$-Learning.** To promote generalization or work with large state spaces, $Q$-learning can be equipped with linear function approximation, where we approximate $q_*(s, a)$ with $x(s, a)^\top w$. Here $x : \mathcal{S} \times \mathcal{A} \to \mathbb{R}^d$ is the feature function that maps a state action pair to a $d$-dimensional feature and $w \in \mathbb{R}^d$ is the learnable weight. Linear $Q$-learning generates the iterates $\{w_t\}$ as

$$A_t \sim \mu_{w_t}(\cdot|S_t), \qquad \text{(linear } Q\text{-learning)}$$
$$\delta_t = R_{t+1} + \gamma \max_{a'} x(S_{t+1}, a')^\top w_t - x(S_t, A_t)^\top w_t,$$
$$w_{t+1} = w_t + \alpha_t \delta_t x(S_t, A_t).$$

Here we have abused $\mu_w$ to also denote the behavior policy used in (linear $Q$-learning), which is defined as

$$\mu_w(a|s) = \frac{\epsilon}{|\mathcal{A}|} + (1 - \epsilon)\frac{\exp(\kappa_w x(s,a)^\top w)}{\sum_b \exp(\kappa_w x(s,b)^\top w)}, \qquad (3)$$

where $\epsilon \in (0, 1]$ and the temperature parameter $\kappa_w$ is defined as

$$\kappa_w = \begin{cases} \frac{\kappa_0}{\|w\|_2} & \|w\|_2 \geq 1 \\ \kappa_0 & \text{otherwise} \end{cases} \qquad (4)$$

with $\kappa_0 > 0$ being a constant. To our knowledge, this behavior policy is first used in Meyn (2024). The softmax approximation of the greedy operator ensures the continuity of $\mu_w$. The adaptive temperature $\kappa_w$ further ensures the Lipschitz continuity of the expected updates of (linear $Q$-learning). Both are key to our finite sample analysis and are discussed in details shortly in the proofs.

## 3. Main Results

**Assumption 3.1.** *The Markov chain $\{S_t\}$ induced by a uniformly random behavior policy is irreducible and aperiodic.*

Notably, since $\epsilon$ is required to be strictly greater than 0, Assumption 3.1 ensures that for any $w$ and any $q$, the Markov chain induced by $\mu_w$ and $\mu_q$ is also irreducible and aperiodic. This is a common assumption to analyze time-inhomogeneous Markovian noise (Zhang et al., 2022).

**Assumption LR.** *The learning rate is $\alpha_t = \frac{\alpha}{(t+t_0)^{\epsilon_\alpha}}$, where $\epsilon_\alpha \in (0.5, 1]$, $\alpha > 0$, $t_0 > 0$ are constants.*

**Theorem 1** ($L^2$ Convergence Rate of Linear $Q$-Learning).
*Let Assumptions 3.1 and LR hold. Then for sufficiently small $\epsilon$ in (3), sufficiently large $\kappa_0$ in (4), and sufficiently large $t_0$ in $\alpha_t$, there exist some constant $\bar{t}$ such that the iterates $\{w_t\}$ generated by (linear $Q$-learning) satisfy the following statements $\forall t \geq \bar{t}$.*
*(1) When $\epsilon_\alpha = 1$, there exist some constants $B_{1,1}$, $B_{1,2}$, and $B_{1,3}$ such that*

$$\mathbb{E}\left[\|w_t\|_2^2\right] \leq \frac{B_{1,1}}{(t+t_0)^{B_{1,2}\alpha}}\|w_0\|_2^2 + B_{1,3}.$$

*(2) When $\epsilon_\alpha \in (0, 1)$, there exist some constants $B_{1,4}$, $B_{1,5}$ and $B_{1,6}$ such that*

$$\mathbb{E}\left[\|w_t\|_2^2\right] \leq B_{1,4} \exp\left(-\frac{B_{1,5}\alpha}{1-\epsilon_\alpha}(t+t_0)^{1-\epsilon_\alpha}\right)\|w_0\|_2^2 + B_{1,6}.$$

The proof of Theorem 1 is in Section 5.2. The precise constraints on $\kappa_0$ and $\epsilon$ are specified in Lemma 16 in Appendix. The constants $B_{1,3}$ and $B_{1,6}$ depend on $\gamma$, $|\mathcal{S}|$, $|\mathcal{A}|$, $\max_{s,a} |r(s, a)|$, and $\max_{s,a} \|x(s, a)\|_2$, with the detailed dependency specified in Section C.4. Theorem 1 confirms the main claim of the work, with $B_{1,3}$ and $B_{1,6}$ corresponding to the bounded set.

**Theorem 2** ($L^2$ Convergence Rate of Tabular $Q$-Learning).
*Let Assumptions 3.1 and LR hold. Then for sufficiently large $t_0$ in $\alpha_t$, there exist some constant $\bar{t}$ such that the iterates $\{w_t\}$ generated by (tabular $Q$-learning) satisfy the following statements $\forall t \geq \bar{t}$.*
*(1) When $\epsilon_\alpha = 1$, there exist some constants $B_{2,1}$, $B_{2,2}$, and $B_{2,3}$ such that*

$$\mathbb{E}\Big[\|q_t - q_*\|_\infty^2\Big]$$
$$\leq \frac{B_{2,1}}{(t+t_0)^{B_{2,2}\alpha}}\|q_0 - q_*\|_\infty^2 + \frac{B_{2,3}\ln(t+t_0)}{(t+t_0)^{\min(1,B_{2,2}\alpha)}}.$$

*(2) When $\epsilon_\alpha \in (0.5, 1)$, for any $\epsilon'_\alpha \in (0.5, \epsilon_\alpha)$, there exist some constants $B_{2,4}$, $B_{2,5}$, and $B_{2,6}$ such that*

$$\mathbb{E}\Big[\|q_t - q_*\|_\infty^2\Big]$$
$$\leq B_{2,4}\exp\Big(-\frac{B_{2,5}\alpha}{1-\epsilon_\alpha}(t+t_0)^{1-\epsilon_\alpha}\Big)\|q_0 - q_*\|_\infty^2$$
$$+ B_{2,6}(t+t_0)^{1-2\epsilon'_\alpha}.$$

The proof of Theorem 2 is in Section 5.3. The constants $B_{2,4}$ and $B_{2,6}$ depends on $\epsilon_\alpha$ and $\epsilon'_\alpha$. Comparing Theorems 1 & 2, it is now clear that (tabular $Q$-learning) converges to the optimal action value function while (linear $Q$-learning) converges only to a bounded set, despite the (linear $Q$-learning) adopts a more complicated behavior policy with an adaptive temperature. We note that the softmax function in (3) and (2) can be replaced by any Lipschitz continuous function. We use the softmax function here because a softmax policy is a commonly used approximation for the greedy policy.

**Stochastic Approximation.** We now present a general stochastic approximation result, which will be used to prove Theorems 1 and 2. In particular, we consider a general iterative update rule for a weight vector $w \in \mathbb{R}^d$, driven by a stochastic process $\{Y_t\}$ evolving in a finite space $\mathcal{Y}$:

$$w_{t+1} = w_t + \alpha_t H(w_t, Y_{t+1}), \quad \text{(SA)}$$

where $H : \mathbb{R}^d \times \mathcal{Y} \to \mathbb{R}^d$ defines the incremental update. The key difficulty in our analysis results from the fact that we allow $\{Y_t\}$ to be a time-inhomogeneous Markov chain with transition functions controlled by $\{w_t\}$.

**Assumption A1.** *There exists a family of parameterized transition matrices $\Lambda_P \doteq \big\{ P_w \in \mathbb{R}^{|\mathcal{Y}|\times|\mathcal{Y}|} | w \in \mathbb{R}^d \big\}$ such that $\Pr(Y_{t+1} \mid Y_t) = P_{w_t}(Y_t, Y_{t+1})$. Furthermore, let $\bar{\Lambda}_P$ denote the closure of $\Lambda_P$. Then for any $P \in \bar{\Lambda}_P$, the time-homogeneous Markov chain induced by $P$ is irreducible and aperiodic.*

Assumption A1 is a standard uniform ergodicity condition (cf. Marbach & Tsitsiklis (2001) and Assumption 3.2 in Zhang et al. (2022)). It is readily satisfied in our framework, and we provide its formal verification in Section 5.2

and 5.3. Specifically, our Assumption 3.1, combined with the use of an $\epsilon$-softmax behavior policy where $\epsilon > 0$, ensures that any policy $P_w \in \Lambda_P$ induces an irreducible and aperiodic Markov chain. Consequently, Assumption A1 ensures that for any $w$, the Markov chain induced by $P_w$ has a unique stationary distribution, which we denote as $d_{\mathcal{Y},w}$. This allows us to define the expected update as $h(w) \doteq \mathbb{E}_{y \sim d_{\mathcal{Y},w}}[H(w, y)]$. One important implication of uniform ergodicity is uniform mixing, which plays a key role in our analysis in Section 5.1. We next present a few assumptions about Lipschitz continuity.

**Assumption A2.** *There exists a constant $C_{A2}$ such that*

$$\|H(w_1, y) - H(w_2, y)\| \leq C_{A2}\|w_1 - w_2\| \quad \forall w_1, w_2, y.$$

**Assumption A3.** *There exists a constant $C_{A3}$ such that*

$$\|P_{w_1} - P_{w_2}\| \leq \frac{C_{A3}}{1+\|w_1\|+\|w_2\|}\|w_1 - w_2\|,$$
$$\|h(w_1) - h(w_2)\| \leq C_{A3}\|w_1 - w_2\| \quad \forall w_1, w_2.$$

**Assumption A3′.** *For any given $w_0$, there exists a constant $U_{A3'}$ such that the iterates $\{w_t\}$ generated by (SA) satisfy*

$$\sup_t \|w_t\| < U_{A3'} \quad a.s.$$

*Furthermore, there exist a constant $C_{A3'}$ such that*

$$\|P_{w_1} - P_{w_2}\| \leq C_{A3'}\|w_1 - w_2\| \quad \forall w_1, w_2,$$

*and for any $w_1, w_2$ satisfying $\max\{\|w_1\|, \|w_2\|\} \leq U_{A3'}$,*

$$\|h(w_1) - h(w_2)\| \leq C_{A3'}\|w_1 - w_2\|.$$

Notably, Assumption A3 uses a stronger Lipschitz condition for $P_w$ than Assumption A3′. Moreover, Assumption A3 requires $h(w)$ to be Lipschitz continuous on $\mathbb{R}^d$ while Assumption A3′ requires $h(w)$ to be Lipschitz continuous only on compact subsets. The price that Assumption A3′ pays to weaken Assumption A3 is an additional assumption that the iterates are bounded by a constant almost surely. The particular multiplier $1/(1 + \|w_1\| + \|w_2\|)$ is inspired by Assumption 5.1(3) of Konda (2002) for analyzing actor critic algorithms. We will use Assumption A3 for (linear $Q$-learning) and Assumption A3′ for (tabular $Q$-learning). Let $w_{\text{ref}}$ be any fixed vector in $\mathbb{R}^d$. We now introduce

$$L(w) = \tfrac{1}{2}\|w - w_{\text{ref}}\|^2 \quad \text{(5)}$$

to present our general results regarding $\{w_t\}$ in (SA).

**Theorem 3.** *Let Assumptions A1, A2, and LR hold. Let at least one of Assumptions A3 and A3′ hold. Then there exist some constant $\bar{t}$ and some function $f(t) = \mathcal{O}\Big(\frac{\ln^2(t+t_0+1)}{(t+t_0)^{2\epsilon_\alpha}}\Big)$ such that the iterates $\{w_t\}$ generated by (SA) satisfy $\forall t \geq \bar{t}$,*

$$\mathbb{E}[L(w_{t+1})]$$
$$\leq (1 + f(t))\mathbb{E}[L(w_t)] + \alpha_t \mathbb{E}[\langle \nabla L(w_t), h(w_t)\rangle] + f(t).$$

The proof of Theorem 3 is in Section 5.1. Theorem 3 gives a recursive bound of $L(w_t)$. We will realize $w_{\text{ref}}$ differently and bound $\langle \nabla L(w_t), h(w_t) \rangle$ differently for (linear $Q$-learning) and (tabular $Q$-learning).

## 4. Related Works

Previous analyses of (linear $Q$-learning) typically involve different modifications to the algorithm. The first is a target network (Mnih et al., 2015), where a slowly changing copy of $\{w_t\}$ is stored. This copy is called the target network and is referred to as $\{\bar{w}_t\}$. Then the TD error $\delta_t$ in (linear $Q$-learning) is computed using $\max_{a'} x(S_{t+1}, a')\bar{w}_t$ instead of $\max_{a'} x(S_{t+1}, a')w_t$. The second is a replay buffer (Lin, 1992). After obtaining the tuple $(S_t, A_t, R_{t+1}, S_{t+1})$, intead of applying (linear $Q$-learning) directly, they store the tuple into a buffer, sample some previous tuple $(s, a, r, s')$ from the buffer, and then apply the update based on the sampled tuple. The third is projection, where the update to $w_t$ is modified as $w_{t+1} = \Pi(w_t + \alpha_t \delta_t x(S_t, A_t))$. Here $\Pi$ is a projection operator that projects the weights into some compact set to make sure $\{w_t\}$ is always bounded. The fourth is regularization. One example is to change the update to $w_t$ as $w_{t+1} = w_t + \alpha_t \delta_t x(S_t, A_t) - \alpha_t \nabla \|w_t\|_2^2$, where the last term corresponds to ridge regularization. Such regularization is also used to facilitate boundedness of $\{w_t\}$. Besides the algorithmic modification, strong assumptions on the features $X$ are also sometimes made. For example, Lee & He (2020) assume all column vectors of $X$ are orthogonal to each other and and each elements of $X$ is either 0 or 1. Finally, different assumptions on the data are also used to facilitate analysis. For example, Lee & He (2020); Carvalho et al. (2020); Han-Dong & Donghwan (2024) assume that $(S_t, A_t, R_{t+1}, S_{t+1})$ is sampled in an i.i.d. manner from a fixed distribution. Gopalan & Thoppe (2022) further assume that $(S_t, A_t, R_{t+1}, S_{t+1})$ is sampled from the stationary distribution of the current $\epsilon$-greedy policy, i.e., not only the samples are i.i.d., but also the sampling distribution changes every time step. Melo et al. (2008); Chen et al. (2022) make strong assumptions on the behavior policy. Melo et al. (2008) use a special matrix condition assuming $\Sigma_\mu - \gamma^2 \Sigma_\mu^*(w)$ is positive definite for any $w$. Here $\Sigma_\mu$ is the feature covariance matrix induced by the fixed behavior policy and $\Sigma_\mu^*(w)$ is the feature covariance matrix induced by the greedy policy w.r.t. $w$. Chen et al. (2022) assume $\gamma^2 \mathbb{E}[\max_a(\phi(s,a)^\top w)^2] < \mathbb{E}[(\phi(s,a)^\top w)^2] \forall w$. Since those assumptions are made w.r.t. all possible $w$, for them to hold, one typically needs to ensure that the behavior policy $\mu$ is close to the optimal policy in certain sense. See Section 6 of Chen et al. (2022) for more discussion about this. Similar strong assumptions are also later on used for analyzing $Q$-learning with neural networks (Cai et al., 2023; Xu & Gu, 2020). As can be seen in Table 1, this work is

the first $L^2$ convergence rate of (linear $Q$-learning) without making any algorithmic modification or strong assumptions on the behavior policy. This setting is known to be computationally challenging. Kane et al. (2023) show that finding the optimal policy is NP-hard even if $q_*$ is linear in the given features. This inherent difficulty highlights the importance of establishing convergence to a bounded set, as our work does, especially under minimal assumptions. Despite that $Q$-learning with neural network is also analyzed in previous works, those work do not supercede our work since they also need to make algorithmic modifications. Some of them even need the Bellman completeness assumption, i.e., for any $w$, the vector $\mathcal{T}Xw$ still lies in the column space of $X$.

There is a rich literature in analyzing (tabular $Q$-learning). Early works (Watkins, 1989; Watkins & Dayan, 1992; Jaakkola et al., 1993; Tsitsiklis, 1994; Littman & Szepesvári, 1996) implicitly or explicitly rely on the use of count-based learning rates, where the $\alpha_t$ in (tabular $Q$-learning) is replaced by $\alpha_{\nu(S_t, A_t, t)}$. Here $\nu(s, a, t)$ counts the number of visits to the state action pair $(s, a)$ until time $t$. The count-based learning rate allows them to work with a wide range of behavior policies. Convergence rates are also later on established by Szepesvári (1997); Even-Dar et al. (2003). However, to our knowledge, such count based learning rate is rarely used by practitioners. Recent works (Chen et al., 2021; Li et al., 2024; Zhang & Xie, 2024) are able to remove the count based learning rate. They, however, need to assume that a fixed behavior policy is used. By contrast, practitioners usually prefer an $\epsilon$-greedy policy. In this work, we use an $\epsilon$-softmax policy to approximate the $\epsilon$-greedy policy and establish the $L^2$ convergence rate, without using count based learning rate.

Meyn (2024) is the closest to this work in terms of the results. However, the underlying techniques are dramatically different. Meyn (2024) uses the ODE based analysis that connects the iterates with the trajectories of certain ODEs (Benveniste et al., 1990; Kushner & Yin, 2003; Borkar, 2009; Borkar et al., 2025; Liu et al., 2025). This work relies on bounding the norm of the iterates recursively (Zou et al., 2019; Chen et al., 2021; Zhang et al., 2022).

Previously, Zhang et al. (2023a) demonstrate convergence rates of linear SARSA (Rummery & Niranjan, 1994) to a bounded set. Our analysis is also largely inspired by Zhang et al. (2023a). However, our analysis is more challenging in two aspects. First, Zhang et al. (2023a) use a projection to ensure the weights are bounded while we do not. Second, linear SARSA is an on-policy algorithm, but linear $Q$-learning is an off-policy algorithm.

The key technical challenge of this work lies in the time-inhomogeneous nature of the Markovian noise. Previously, this is often tackled in a two-timescale framework, where the transition function of the Markovian noise is controlled by

a secondary weight, say $\{\theta_t\}$, and $\theta_t$ evolves much slower than $w_t$ (Konda & Tsitsiklis, 1999; Wu et al., 2020; Zhang et al., 2020; 2022). This is recently improved by Olshevsky & Gharesifard (2023); Chen & Zhao (2023) to allow $\theta_t$ and $w_t$ to evolve in the same timescale. Our analysis is more challenging in that we actually have $\theta_t \equiv w_t$. This setting is also previously considered in Zou et al. (2019); Zhang et al. (2023a) but they all rely on a projection operator. By contrast, our (SA) does not have any projection.

# 5. Proofs of the Main Results

## 5.1. Proof of Theorem 3

*Proof.* We start by recalling that we use $\|\cdot\|$ to denote an arbitrary norm where $\|\cdot\|^2$ is smooth. In particular, $\frac{1}{2}\|\cdot\|^2$ is smooth w.r.t. $\|\cdot\|$ for some $L$. In (1), identifying $f(w)$ as $L(w)$, $w'$ as $w_{t+1}$, and $w$ as $w_t$ yields

$$
\begin{aligned}
L(w_{t+1}) \leq & L(w_t) + \alpha_t \langle \nabla L(w_t), H(w_t, Y_{t+1}) \rangle \\
& + \tfrac{L\alpha_t^2}{2} \|H(w_t, Y_{t+1})\|^2 \\
= & L(w_t) + \alpha_t \langle \nabla L(w_t), h(w_t) \rangle \\
& + \alpha_t \langle \nabla L(w_t), H(w_t, Y_{t+1}) - h(w_t) \rangle \\
& + \tfrac{L\alpha_t^2}{2} \|H(w_t, Y_{t+1})\|^2.
\end{aligned}
\tag{6}
$$

We now proceed to bounding the noise term including $H(w_t, Y_{t+1}) - h(w_t)$. To this end, we notice that according to Lemma 1 of Zhang et al. (2022), Assumption A1 implies that the Markov chains in $\Lambda_P$ mix both geometrically and uniformly. In other words, there exist constants $C_0 > 0$ and $\tau \in (0, 1)$, such that

$$
\sup_{w,y} \sum_{y'} |P_w^n(y, y') - d_{\mathcal{Y},w}(y')| \leq C_0 \tau^n. \tag{7}
$$

We then define

$$
\tau_\alpha \doteq \min\{n \geq 0 \mid C_0 \tau^n \leq \alpha\} \tag{8}
$$

to denote the number of steps that the Markov chain needs to mix to an accuracy $\alpha$. This allows us to decompose the noise term as

$$
\begin{aligned}
& \langle \nabla L(w_t), H(w_t, Y_{t+1}) - h(w_t) \rangle \tag{9} \\
= & \underbrace{\langle \nabla L(w_t) - \nabla L(w_{t-\tau_{\alpha_t}}), H(w_t, Y_{t+1}) - h(w_t) \rangle}_{T_1} \\
& + \langle \nabla L(w_{t-\tau_{\alpha_t}}), H(w_t, Y_{t+1}) - H(w_{t-\tau_{\alpha_t}}, Y_{t+1}) \\
& \underbrace{\quad + h(w_{t-\tau_{\alpha_t}}) - h(w_t) \rangle}_{T_2} \\
& + \underbrace{\langle \nabla L(w_{t-\tau_{\alpha_t}}), H(w_{t-\tau_{\alpha_t}}, Y_{t+1}) - h(w_{t-\tau_{\alpha_t}}) \rangle}_{T_3}.
\end{aligned}
$$

Both $T_1$ and $T_2$ can be bounded with Lipschitz continuity. To bound $T_3$, we introduce an auxiliary Markov

chain $\{\tilde{Y}_t\}$ akin to Zou et al. (2019). The chain $\{\tilde{Y}_t\}$ is constructed to be identical to $\{Y_t\}$ up to time step $t - \tau_{\alpha_t}$, after which it evolves independently according to the fixed transition matrix $P_{w_{t-\tau_{\alpha_t}}}$. By contrast, $\{Y_t\}$ continues to evolve according to the changing transition matrix $P_{w_{t-\tau_{\alpha_t}}}, P_{w_{t-\tau_{\alpha_t}+1}}, \cdots$. This choice of $\tau_{\alpha_t}$ ensures that the discrepancy between the two chains is sufficiently small. We now further decompose $T_3$ with this auxiliary chain as

$$
\begin{aligned}
& \langle \nabla L(w_{t-\tau_{\alpha_t}}), H(w_{t-\tau_{\alpha_t}}, Y_{t+1}) - h(w_{t-\tau_{\alpha_t}}) \rangle \tag{10} \\
= & \underbrace{\langle \nabla L(w_{t-\tau_{\alpha_t}}), H(w_{t-\tau_{\alpha_t}}, \tilde{Y}_{t+1}) - h(w_{t-\tau_{\alpha_t}}) \rangle}_{T_{31}} \\
& + \underbrace{\langle \nabla L(w_{t-\tau_{\alpha_t}}), H(w_{t-\tau_{\alpha_t}}, Y_{t+1}) - H(w_{t-\tau_{\alpha_t}}, \tilde{Y}_{t+1}) \rangle}_{T_{32}}.
\end{aligned}
$$

We now bound the terms one by one.

**Lemma 1.** *There exists a constant $C_1$ such that*

$$
T_1 \leq C_1 \alpha_{t-\tau_{\alpha_t}, t-1}(L(w_t) + 1).
$$

The proof is in Section B.1.

**Lemma 2.** *There exists a constant $C_2$ such that*

$$
T_2 \leq C_2 \alpha_{t-\tau_{\alpha_t}, t-1}(L(w_t) + 1).
$$

The proof is in Section B.2.

**Lemma 3.** *There exists a constant $C_3$ such that*

$$
\mathbb{E}[T_{31}] \leq C_3 \alpha_t(\mathbb{E}[L(w_t)] + 1).
$$

The proof is in Section B.3.

**Lemma 4.** *There exists a constant $C_4$ such that*

$$
\mathbb{E}[T_{32}] \leq C_4 \alpha_{t-\tau_{\alpha_t}, t-1} \ln(t + t_0 + 1)(\mathbb{E}[L(w_t)] + 1).
$$

The proof is in Section B.4.

**Remark 1.** We prove Lemma 4 considering Assumption A3 or A3′ in two cases. Under Assumption A3′, our proof is similar to those in Zou et al. (2019); Zhang et al. (2022). The technical novelty here lies in the proof under Assumption A3. The proof under Assumption A3 involves bounding the term $\|P_{w_t} - P_{w_{t-\alpha_t}}\|$. In Zou et al. (2019), a similar term is bounded by $\|w_t - w_{t-\alpha_t}\|$ via the standard Lipschitz continuity of $P_w$ (cf. Assumption A3′). Zou et al. (2019) further rely on a projection operator to bound $\|w_t - w_{t-\alpha_t}\|$ directly. However, (SA) does not have such a projection operator. In Zhang et al. (2022), a similar term is bounded under the assumption that the transition

matrix is controlled by another set of weights $\{\theta_t\}$, which involve much slower than $\{w_t\}$. Their bound is made possible essentially because of this two-timescale setup. However, (SA) only has a single timescale where the transition matrix evolves as fast as $\{w_t\}$. We instead use a stronger form of Lipschitz continuity in Assumption A3.

Assembling the bounds in the above lemmas to (9) and further to (6), we complete the proof of Theorem 3. The detailed steps are presented in Section B.5. □

In the following sections, we first map the general update (SA) to (linear $Q$-learning) and (tabular $Q$-learning) by defining $H(w, y)$, $h(w)$, $L(w)$, and the norm $\|\cdot\|$ properly. We then bound the remaining term $\langle \nabla L(w_t), h(w_t) \rangle$ in Theorem 3 separately to complete the proof.

### 5.2. Proof of Theorem 1

*Proof.* We first rewrite (linear $Q$-learning) in the form of (SA). To this end, we define $Y_{t+1} \doteq (S_t, A_t, S_{t+1})$, which evolves in a finite space

$$\mathcal{Y} \doteq \{(s \in \mathcal{S}, a \in \mathcal{A}, s' \in \mathcal{S}) \mid p(s'|s, a) > 0\}.$$

We further define

$$H(w, y) \tag{11}$$
$$\doteq (r(s, a) + \gamma \max_{a'} x(s', a')^\top w - x(s, a)^\top w) x(s, a).$$

where we have used $y \doteq (s, a, s')$. We now proceed to verify the assumptions of Theorem 3. We identify the norm used in Theorem 3 as the $\ell_2$ norm $\|\cdot\|_2$. For Assumption A1, we define $P_w$ as

$$P_w[(s_0, a_0, s_0'), (s_1, a_1, s_1')]$$
$$\doteq \mu_w(a_1|s_1) p(s_1'|s_1, a_1) \mathbb{I}_{s_0'=s_1},$$

where $\mathbb{I}$ is the indicator funciton. Then it is easy to see that

$$\Pr(Y_{t+1}|Y_t) = \mu_{w_t}(A_t|S_t) p(S_{t+1}|S_t, A_t)$$
$$= P_{w_t}[(S_{t-1}, A_{t-1}, S_t), (S_t, A_t, S_{t+1})]$$
$$= P_{w_t}[Y_t, Y_{t+1}].$$

Thanks to Assumption 3.1 and the fact $\epsilon > 0$ in (3), it is easy to see that for any $P \in \bar{\Lambda}_P$, $P$ induces an irreducible and aperiodic chain in $\mathcal{Y}$. Assumption A1 is then verified. Assumption A2 trivially holds according to the definition of $H(w, y)$ in (11) since max is Lipschitz and $\mathcal{Y}$ is finite.

For Assumption A3, the first half is trivially implied by the following lemma.

**Lemma 5.** *There exists a constant $C_5$ such that*

$$|\mu_{w_1}(a|s) - \mu_{w_2}(a|s)| \leq \frac{C_5 \|w_1 - w_2\|_2}{\|w_1\|_2 + \|w_2\|_2 + 1} \quad \forall w_1, w_2, s, a.$$

The proof is in Section C.1, where the adaptive temperature $\kappa_w$ plays a key role. For the second half of Assumption A3, we use $d_{\mu_w} \in \mathbb{R}^{|\mathcal{S}||\mathcal{A}|}$ to denote the stationary state-action distribution induced by the policy $\mu_w$. Then it is easy to see that $d_{\mathcal{Y},w}(s, a, s') = d_{\mu_w}(s, a)p(s'|s, a)$. Then it can be computed that

$$h(w) = \mathbb{E}_{y \sim d_{\mathcal{Y},w}}[H(w, y)] = A(w)w + b(w), \tag{12}$$

where

$$A(w) \doteq X^\top D_{\mu_w}(\gamma P_{\pi_w} - I)X, \tag{13}$$
$$b(w) \doteq X^\top D_{\mu_w} r, \tag{14}$$

Here, we use $X \in \mathbb{R}^{|\mathcal{S}||\mathcal{A}| \times d}$ to denote the feature matrix, the $(s, a)$-indexed row of which is $x(s, a)^\top$. We use $D_{\mu_w} \in \mathbb{R}^{|\mathcal{S}||\mathcal{A}| \times |\mathcal{S}||\mathcal{A}|}$ to denote a diagonal matrix whose diagonal is $d_{\mu_w}$. We use $\pi_w$ to denote the greedy policy, i.e.,

$$\pi_w(a|s) = \begin{cases} 1 & \text{if} \quad a = \arg\max_b x(s, b)^\top w \\ 0 & \text{otherwise} \end{cases},$$

where tie-breaking in $\arg\max$ can be done through any fixed and deterministic prodecure. We recall that $P_{\pi_w} \in \mathbb{R}^{|\mathcal{S}||\mathcal{A}| \times |\mathcal{S}||\mathcal{A}|}$ denotes the state-action transition matrix of the policy $\pi_w$. We then have

**Lemma 6.** *There exists a constant $C_6$ such that*

$$\|h(w_1) - h(w_2)\|_2 \leq C_6 \|w_1 - w_2\|_2 \quad \forall w_1, w_2.$$

The proof is Section C.2. Assumption A3 is then verified.

**Remark 2.** To prove the above lemma, we need to bound a term involving $\|D_{\mu_{w_1}} - D_{\mu_{w_2}}\|_2 \|w_1\|_2$, for which we rely on the factor $1/(\|w_1\|_2 + \|w_2\|_2 + 1)$ in Lemma 5 to cancel the multiplicative term $\|w_1\|_2$. Notably, as discussed at the end of Section 2, an $\epsilon$-greedy policy, due to its discontinuities w.r.t. $w$, will invalidate Lemma 5 and consequently prevent the establishment of Lemma 6. Furthermore, despite that $\pi_w$ is not continuous, $P_{\pi_w} X w$ is Lipschitz continuous, thanks to the adaptive temperature $\kappa_w$.

We now identify $w_{\text{ref}} = 0$ and thus have $L(w) = \frac{1}{2}\|w\|_2^2$. Invoking Theorem 3 then yields

$$\mathbb{E}[L(w_{t+1})] \tag{15}$$
$$\leq (1 + f(t))\mathbb{E}[L(w_t)] + \alpha_t \mathbb{E}[\langle \nabla L(w_t), h(w_t) \rangle] + f(t).$$

Since $\nabla L(w_t) = w_t$, we now bound the inner product term of the RHS with the following lemma.

**Lemma 7.** *There exist constants $\beta > 0$ and $C_7 > 0$ such that*

$$\langle w, h(w) \rangle \leq -\beta \|w\|_2^2 + C_7 \|w\|_2 \quad \forall w.$$

The proof is in Section C.3, which is an extension of Lemma A.9 of Meyn (2024). Notably, this lemma is the place where we require sufficiently large $\kappa_0$ and sufficiently small $\epsilon$ in (3). Plugging the bound in Lemma 7 back to (15) yields

$$\mathbb{E}[L(w_{t+1})] \leq (1 - \beta\alpha_t + f(t))\mathbb{E}[L(w_t)] + \mathcal{O}(\alpha_t).$$

Since $f(t)$ is dominated by $\alpha_t$, it can be seen that $\mathbb{E}[L(w_t)]$ remains bounded as $t \to \infty$. By telescoping the above inequality, we can also obtain an explicit convergence rate of $\mathbb{E}[L(w_t)]$ to a bounded set. All those details are in Section C.4, which completes the proof. □

### 5.3. Proof of Theorem 2

*Proof.* It is obvious that (tabular $Q$-learning) is a special case of (linear $Q$-learning) with $X$ identified as the identity matrix $I$. Most of the proofs here are similar to Section 5.3. We, therefore, focus on the different part.

Assumption A1 can be similarly verified with

$$
\begin{aligned}
&P_q[(s_0, a_0, s_0'), (s_1, a_1, s_1')] \\
&\doteq \mu_q(a_1|s_1)p(s_1'|s_1, a_1)\mathbb{I}_{s_0'=s_1},
\end{aligned}
$$

where $\mu_q$ is defined in (2). For the other assumptions, by using $X = I$ in (12), we obtain

$$
\begin{aligned}
h(q) &= D_{\mu_q}(\gamma P_{\pi_q} - I)q + D_{\mu_q}r \\
&= D_{\mu_q}(\mathcal{T}q - q) \\
&= \mathcal{T}'q - q, \tag{16}
\end{aligned}
$$

where we have defined the operator $\mathcal{T}'$ as

$$\mathcal{T}'q \doteq D_{\mu_q}(\mathcal{T}q - q) + q.$$

Here we call $\mathcal{T}'$ the weighted Bellman optimality operator. The operator $\mathcal{T}'$ is highly nonlinear since $D_{\mu_q}$ depends on the stationary distribution of the chain induced by the policy $\mu_q$. This operator $\mathcal{T}'$ is unlikely to be a contraction. The key insight that we rely on here is that $\mathcal{T}'$ is still a pseudo-contraction w.r.t. the infinity norm $\|\cdot\|_\infty$.

**Lemma 8.** *Let Assumption 3.1 hold. For any $\epsilon > 0$, the weighted Bellman operator $\mathcal{T}'$ is a pseudo-contraction and $q_*$ is the unique fixed point. In other words, for any $q$, it holds that*

$$\|\mathcal{T}'q - q_*\|_\infty \leq \beta_m\|q - q_*\|_\infty,$$

*where $\beta_m \doteq 1 - (1 - \gamma)\inf_{q,s,a} d_{\mu_q}(s, a) \in (0, 1)$.*

The proof is in Section D.1.

**Remark 3.** This lemma is, to our knowledge, the first time that this pseudo contraction property is identified. The inf is strictly greater than 0 because $\epsilon > 0$.

The pseudo contraction property motivates us to identify the $\|\cdot\|$ in Theorem 3 as $\|\cdot\|_\infty$. Unfortunately, $\|\cdot\|_\infty^2$ is not smooth. We then follow Chen et al. (2021) and use the Moreau envelope of $\|\cdot\|_\infty^2$ defined as

$$M(q) \doteq \inf_{u \in \mathbb{R}^d} \left\{ \frac{1}{2}\|u\|_\infty^2 + \frac{1}{2\xi}\|q - u\|_2^2 \right\},$$

where $\xi > 0$ is a constant to be tuned. Due to the equivalence between norms, there exist positive constants $l_{it}$ and $u_{it}$ such that $l_{it}\|q\|_2 \leq \|q\|_\infty \leq u_{it}\|q\|_2$. The properties of $M(q)$ are summarized below.

**Lemma 9.** *(Proposition A.1 and Section A.2 of Chen et al. (2021))*

(i). *$M(q)$ is $\frac{2}{\xi}$-smooth w.r.t. $\|\cdot\|_2$.*

(ii). *There exists a norm $\|\cdot\|_m$ such that $M(q) = \frac{1}{2}\|q\|_m^2$.*

(iii). *Define $l_{im} = \sqrt{(1 + \xi l_{it}^2)}$, $u_{im} = \sqrt{(1 + \xi u_{it}^2)}$, then $l_{im}\|q\|_m \leq \|q\|_\infty \leq u_{im}\|q\|_m$.*

(iv). *$\langle \nabla M(q), q' \rangle \leq \|q\|_m\|q'\|_m$, $\langle \nabla M(q), q \rangle \geq \|q\|_m^2$.*

We then identify the norm $\|\cdot\|$ in Theorem 3 as $\|\cdot\|_m$ and further identify $w_{\text{ref}}$ as $q_*$. As a result, we have realized $L(w)$ as

$$L(q) = \frac{1}{2}\|q - q_*\|_m^2 = M(q - q_*).$$

Assumption A2 again trivially holds. We now proceed to verify Assumption A3$'$. The boundedness of $\{q_t\}$ can be easily obtained via induction (cf. Gosavi (2006)).

**Lemma 10.** *For any $q_0$, there exists a constant $C_{10}$ such that $\sup_t \|q_t\|_m \leq C_{10}$ a.s.*

The proof is in Section D.2. The Lipschitz continuity of $P_q$ follows directly from the Lipschitz continuity of $\mu_q$ in (2), which is a direct result from the fact that the softmax function is Lipschitz continuous. The Lipschitz continuity of $h(q)$ on a compact set is also simpler to prove than Lemma 6.

**Lemma 11.** *There exists a constant $C_{11}$ such that for any $q_1, q_2$ satisfying $\max\{\|q_1\|_m, \|q_2\|_m\} < C_{10}$, it holds that*

$$\|h(q_1) - h(q_2)\|_m \leq C_{11}\|q_1 - q_2\|_m.$$

The proof is in Section D.3. Assumption A3$'$ is now verified. Invoking Theorem 3 then yields

$$
\begin{aligned}
&\mathbb{E}[L(q_{t+1})] \tag{17} \\
&\leq (1 + f(t))\mathbb{E}[L(q_t)] + \alpha_t\mathbb{E}[\langle \nabla L(q_t), h(q_t) \rangle] + f(t).
\end{aligned}
$$

The pseudo-contraction property of $\mathcal{T}'$ allows us to bound the inner product in the RHS as below.

**Lemma 12.** *There exists a positive constant $C_{12}$ such that*

$$\langle \nabla L(q_t), h(q_t) \rangle \leq -C_{12} L(q_t) \; \forall t.$$

The proof is in Section D.4. Plugging the bound in Lemma 12 back to (17), we get

$$\mathbb{E}[L(q_{t+1})] \leq (1 - C_{12}\alpha_t + f(t))\mathbb{E}[L(q_t)] + f(t).$$

Since $f(t)$ is dominated by $\alpha_t$, telescoping the above inequality then generates the desired convergence rate. The detailed steps are in Section D.5, which completes the proof. □

## 6. Experiments

We test unmodified linear $Q$-learning (without target networks, experience replay, weight projection, or regularization) on Baird's counterexample, a challenging benchmark for temporal difference methods, especially $Q$-learning (See Chapter 11 of Sutton & Barto (2018)). For clearer experimental results, we use a constant learning rate $\alpha = 0.1$, we also select $\kappa_0 = 100$ and $\epsilon = 0.1$.

In Baird's counterexample, we have seven states, two actions, zero reward $r(s, a) = 0$ for all state-action pairs, and discount factor $\gamma = 0.99$. The environment uses a specific feature representation and all transitions lead to state 7 with probability 1, regardless of action.

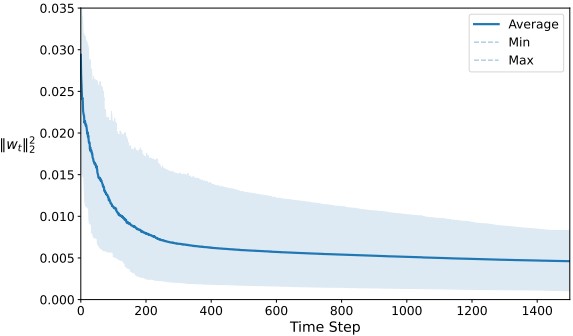

*Figure 1.* Convergence of (linear $Q$-learning) with $\gamma = 0.99, \alpha = 0.1$. The graph shows the evolution of $\|w_t\|_2^2$ over time steps, demonstrating stable convergence behavior. The blue line represents the average of the squared $L^2$ norm of weights over 10 independent runs, and the shaded area indicates the range between minimum and maximum values.

Figure 1 shows the evolution of $\|w_t\|^2$ over 1500 iterations. We observed that the weights of unmodified linear $Q$-learning remain stably bounded throughout training, supporting our theoretical findings. Appendix E provides comparisons with versions incorporating other modifications.

## 7. Conclusion

This paper establishes novel $L^2$ convergence rates for both linear and tabular $Q$-learning. A key novelty of the result is that we allow the behavior policy to depend on the current action value estimation, without making any algorithmic modification or strong assumptions. Technically, such a behavior policy is hard to analyze because it brings in time-inhomogeneous Markovian noise, for which we provide Theorem 3 as a general tool. A possible future work is to characterize $Q$-learning with such a behavior policy from other aspects, e.g., almost sure convergence rates, high probability concentration, and $L^p$ convergence rates, following recent works like Chen et al. (2025); Qian et al. (2024).

## Impact Statement

This paper presents work whose goal is to advance the field of Machine Learning. There are many potential societal consequences of our work, none which we feel must be specifically highlighted here.

## Acknowledgments and Disclosure of Funding

This work is supported in part by the US National Science Foundation under grants III-2128019 and SLES-2331904.

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

# A. Auxiliary Lemmas and Notations

**Lemma 13** (Definition 5.1 and Lemma 5.7 of Beck (2017))**.** *The following statements about a differentiable function $f(x)$ are equivalent:*

  *(i)* $f(x)$ *is L-smooth w.r.t. a norm* $\|\cdot\|$.

  *(ii)* $\|\nabla f(x) - \nabla f(y)\|_* \le L\|x - y\|$.

  *(iii)* $|f(y) - f(x) - \langle \nabla f(x), y - x \rangle| \le \frac{L}{2}\|x - y\|^2$.

**Lemma 14** (Discrete Gronwall Inequality, Lemma 8 in Section 11.2 of Borkar (2009))**.** *For non-negative real sequences* $\{x_n, n \ge 0\}$ *and* $\{a_n, n \ge 0\}$ *and scalar* $L \ge 0$, *it holds*

$$x_{n+1} \le C + L\textstyle\sum_{i=0}^n a_i x_i \quad \forall n \implies x_{n+1} \le (C + x_0)\exp(L\textstyle\sum_{i=0}^n a_i) \quad \forall n.$$

**Lemma 15** (Lemma 9 of Zhang et al. (2021))**.** *Let* $\mathcal{U}$ *be a set of policies and* $\Lambda_P \doteq \big\{ P_\mu \in \mathbb{R}^{|\mathcal{S}||\mathcal{A}| \times |\mathcal{S}||\mathcal{A}|} \mid \mu \in \mathcal{U} \big\}$ *be the set of induced state action transition matrices. Let* $\bar{\Lambda}_P$ *be the closure of* $\Lambda_P$. *Assume for each* $P \in \bar{\Lambda}_P$, *the chain on* $\mathcal{S} \times \mathcal{A}$ *induced by* $P$ *is irreducible and aperiodic. Let* $d_{P_\mu}$ *be the stationary distribution of the chain induced by* $P_\mu$. *Then* $d_{P_\mu}$ *is Lipschitz continuous in* $\mu$ *on* $\mathcal{U}$.

Recall the definition of $A(w)$ in (13). We have

**Lemma 16** (Lemma A.9 of Meyn (2024))**.** *There exists a positive constant* $\bar{\kappa}_0$ *such that for* $\kappa_0 > \bar{\kappa}_0$ *and* $\epsilon < (1 - \gamma)^2/\big[(1 - \gamma)^2 + \gamma^2\big]$, *there exists a positive constant* $\beta > 0$ *such that* $w^\top A(w)w \le -\beta\|w\|_2^2$ *holds for all* $\|w\|_2 \ge 1$. *To be more specific,*

$$\beta = \Big[(1 - \gamma) - \epsilon\gamma\sqrt{\epsilon^{-1} + (1 - \epsilon)^{-1}}\Big]\lambda_{\min}(X^\top D_{\mu_w}X) - \gamma(1 - \epsilon)\frac{\log(|\mathcal{A}|)}{\kappa_0}\sqrt{\lambda_{\max}(X^\top D_{\mu_w}X)}$$

*where* $\lambda_{\min}(\cdot)$ *and* $\lambda_{\max}(\cdot)$ *denote the minimal and maximal eigenvalue of the matrix, respectively.*

We define shorthand

$$\alpha_{i,j} \doteq \sum_{t=i}^j \alpha_t, \quad C_x \doteq \max_{s,a}\|x(s,a)\|_2, \quad C_r \doteq \|r\|_\infty, \quad C_{\mathrm{ref}} \doteq \|w_{\mathrm{ref}}\|.$$

We use $\mathcal{F}_t = \sigma(w_0, Y_1, ..., Y_t)$ to denote the filtration representing the history up to time step $t$. Recall the definition of $\tau_\alpha$ in (8). We have

**Lemma 17** (Lemma 11 of Zhang et al. (2022))**.** *For sufficiently large* $t_0$, *it holds that*

$$\tau_{\alpha_t} = \mathcal{O}(\log(t + t_0)), \quad \alpha_{t - \tau_{\alpha_t}, t - 1} = \mathcal{O}\left(\frac{\log(t + t_0)}{(t + t_0)^{\epsilon_\alpha}}\right).$$

Lemma 17 ensures that there exists some $\bar{t} > 0$ (depending on $t_0$) such that for all $t \ge \bar{t}$, it holds that $t \ge \tau_{\alpha_t}$. Throughout the appendix, we always assume $t_0$ is sufficiently large and $t \ge \bar{t}$. We will refine (i.e., increase) $\bar{t}$ along the proof when necessary.

# B. Proofs in Section 5.1

**Lemma 18.** *There exists a constant* $C_{18}$ *such that for any* $w, y, t$,

$$\|H(w,y)\| \le C_{18}(\|w\| + 1),$$
$$\|h(w_t)\| \le C_{18}(\|w_t\| + 1).$$

*Proof.* Recall Assumption A2, we have

$$\|H(w,y) - H(0,y)\| \le C_{A2}\|w\| \quad \forall w, y.$$

According to the triangle inequality, we can obtain

$$\|H(w, y)\| \le \|H(w, y) - H(0, y)\| + \|H(0, y)\|.$$

Therefore, we can further obtain

$$\|H(w, y)\| \le C_{A2}\|w\| + \|H(0, y)\|.$$

Thus, choosing $C_{18,1} \doteq \max_y \{C_{A2}, H(0, y)\}$ completes the proof for bounding $H(w, y)$. Similar result can be obtained for $h(w)$ under Assumption A3 by choosing $C_{18,2} \doteq \max\{C_{A3}, h(0)\}$ and under Assumption A3′ by choosing $C_{18,2} \doteq \max\{C_{A3'}, h(0)\}$. Therefore, we can choose $C_{18} \doteq \max\{C_{18,1}, C_{18,2}\}$. □

**Lemma 19.** *For sufficiently large $t_0$, there exists a constant $C_{19}$ such that the following statement holds. For any $t \ge \bar{t}$ and any $i \in [t - \tau_{\alpha_t}, t]$, it holds that*

$$\left\|w_i - w_{t - \tau_{\alpha_t}}\right\| \le C_{19}\alpha_{t - \tau_{\alpha_t}, i-1}(\|w_i\| + 1), \tag{18}$$

$$\left\|w_i - w_{t - \tau_{\alpha_t}}\right\| \le C_{19}\alpha_{t - \tau_{\alpha_t}, i-1}(\|w_i - w_{ref}\| + 1), \tag{19}$$

$$\left\|w_{t - \tau_{\alpha_t}}\right\| \le C_{19}(\|w_t - w_{ref}\| + 1). \tag{20}$$

*Proof.* In this proof, to simplify notations, we define shorthand $t_1 \doteq t - \tau_{\alpha_t}$. Given Lemma 17, we can select a sufficiently large $t_0$ such that for any $t \ge \bar{t}$,

$$\exp\left(C_{18}\alpha_{t - \tau_{\alpha_t}t-1}\right) < 3,$$

$$C_{18}\alpha_{t - \tau_{\alpha_t}t-1} < \frac{1}{6}.$$

We then bound $\|w_i - w_{t_1}\|$ as

$$\begin{aligned}
\|w_i - w_{t_1}\| &= \sum_{k=t_1}^{i-1} \|\alpha_k H(w_k, Y_{k+1})\| \\
&\le \sum_{k=t_1}^{i-1} \alpha_k C_{18}(\|w_k - w_{t_1}\| + \|w_{t_1}\| + 1) \\
&\le \sum_{k=t_1}^{i-1} \alpha_k C_{18}(\|w_{t_1}\| + 1) + \sum_{k=t_1}^{i-1} \alpha_k C_{18}(\|w_k - w_{t_1}\|) \\
&\le C_{18}\alpha_{t_1, i-1}(\|w_{t_1}\| + 1) \exp(C_{18}\alpha_{t_1, t-1}). \quad \text{(Lemma 14)}
\end{aligned}$$

We then have

$$\begin{aligned}
\|w_i - w_{t_1}\| &\le C_{18}\alpha_{t_1, i-1}(\|w_i - w_{t_1}\| + \|w_i\| + 1)\exp(C_{18}\alpha_{t_1, t-1}) \\
&\le \frac{1}{2}\|w_i - w_{t_1}\| + C_{19,2}\alpha_{t_1, i-1}(\|w_i\| + 1),
\end{aligned}$$

where we have defined $C_{19,2} \doteq 3C_{18}$. Thus, we have

$$\|w_i - w_{t_1}\| \le 2C_{19,2}\alpha_{t_1, i-1}(\|w_i\| + 1),$$

which gives a proof of (18). Furthermore, we obtain (19) as

$$\begin{aligned}
\|w_i - w_{t_1}\| &\le 2C_{19,2}\alpha_{t_1, i-1}(\|w_i\| + 1) \\
&\le 2C_{19,2}\alpha_{t_1, i-1}(\|w_i - w_{\text{ref}}\| + \|w_{\text{ref}}\| + 1) \\
&\le C_{19,3}\alpha_{t_1, i-1}(\|w_i - w_{\text{ref}}\| + 1).
\end{aligned}$$

For (20), taking $i = t$ in the above inequality yields

$$\|w_{t_1}\| - \|w_t\| \le \|w_t - w_{t_1}\| \le C_{19,3}\alpha_{t_1, t-1}(\|w_t - w_{\text{ref}}\| + 1).$$

That is

$$\|w_{t_1}\| \le C_{19,3}\alpha_{t_1,t-1}(\|w_t - w_{\text{ref}}\| + 1) + \|w_t - w_{\text{ref}}\| + \|w_{\text{ref}}\|$$
$$\le C_{19,4}(\|w_t - w_{\text{ref}}\| + 1) + \|w_t - w_{\text{ref}}\| + \|w_{\text{ref}}\|,$$

which completes the proof. $\square$

### B.1. Proof of Lemma 1

*Proof.* According to (9),

$$T_1 = \langle \nabla L(w_t) - \nabla L(w_{t-\tau_{\alpha_t}}), H(w_t, Y_{t+1}) - h(w_t) \rangle$$
$$\le \|\nabla L(w_t) - \nabla L(w_{t-\tau_{\alpha_t}})\|_* \cdot \|H(w_t, Y_{t+1}) - h(w_t)\|.$$

For the first term,

$$\|\nabla L(w_t) - \nabla L(w_{t-\tau_{\alpha_t}})\|_*$$
$$\le L\|w_t - w_{t-\tau_{\alpha_t}}\| \quad \text{(By (5) and Lemma 13)}$$
$$\le LC_{19}\alpha_{t-\tau_{\alpha_t},t-1}(\|w_t - w_{\text{ref}}\| + 1) \quad \text{(Lemma 19)}.$$

For the second term,

$$\|H(w_t, Y_{t+1}) - h(w_t)\| \le C_{18}(\|w_t\| + 1) + C_{18}(\|w_t\| + 1) \le 2C_{18}(\|w_t - w_{\text{ref}}\| + C_{\text{ref}} + 1).$$

Combining the two inequalities yields

$$\langle \nabla L(w_t) - \nabla L(w_{t-\tau_{\alpha_t}}), H(w_t, Y_{t+1}) - h(w_t) \rangle$$
$$\le LC_{19}\alpha_{t-\tau_{\alpha_t},t-1}(\|w_t - w_{\text{ref}}\| + C_{\text{ref}} + 1) \cdot 2C_{18}(\|w_t - w_{\text{ref}}\| + C_{\text{ref}} + 1)$$
$$\le C_{1,1}\alpha_{t-\tau_{\alpha_t},t-1}(\|w_t - w_{\text{ref}}\| + 1)^2$$
$$\le 2C_{1,1}\alpha_{t-\tau_{\alpha_t},t-1}(\|w_t - w_{\text{ref}}\|^2 + 1).$$

Choosing $C_1 \doteq 4C_{1,1}$ then completes the proof. $\square$

### B.2. Proof of Lemma 2

*Proof.* According to (9),

$$T_2 = \langle \nabla L(w_{t-\tau_{\alpha_t}}), H(w_t, Y_{t+1}) - H(w_{t-\tau_{\alpha_t}}, Y_{t+1}) + h(w_{t-\tau_{\alpha_t}}) - h(w_t) \rangle$$
$$\le \|\nabla L(w_{t-\tau_{\alpha_t}})\|_* \cdot \|H(w_t, Y_{t+1}) - H(w_{t-\tau_{\alpha_t}}, Y_{t+1}) + h(w_{t-\tau_{\alpha_t}}) - h(w_t)\|.$$

For the first term, it is trivial to see $\nabla L(w_{\text{ref}}) = 0$. We can then obtain

$$\|\nabla L(w_{t-\tau_{\alpha_t}})\|_*$$
$$= \|\nabla L(w_{t-\tau_{\alpha_t}}) - \nabla L(w_{\text{ref}})\|_*$$
$$\le L\|w_{t-\tau_{\alpha_t}} - w_{\text{ref}}\| \quad \text{(Lemma 13)}$$
$$\le L(\|w_{t-\tau_{\alpha_t}} - w_t\| + \|w_t - w_{\text{ref}}\|)$$

Recall for $\bar{t}$ sufficiently large, we have $C_{19}\alpha_{t-\tau_{\alpha_t},t-1} < 1$. Applying Lemma 19 then yields

$$\|\nabla L(w_{t-\tau_{\alpha_t}})\|_* \le L(\|w_{t-\tau_{\alpha_t}} - w_t\| + \|w_t - w_{\text{ref}}\|)$$
$$\le L(\|w_t - w_{\text{ref}}\| + 1 + \|w_t - w_{\text{ref}}\|)$$
$$\le L(2\|w_t - w_{\text{ref}}\| + 1). \tag{21}$$

For the second term,

$$\big\|H(w_t, Y_{t+1}) - H(w_{t-\tau_{\alpha_t}}, Y_{t+1}) + h(w_{t-\tau_{\alpha_t}}) - h(w_t)\big\|$$
$$\leq (C_{A2} + \max\{C_{A3}, C_{A3'}\})\big\|w_{t-\tau_{\alpha_t}} - w_t\big\|$$
$$\leq C_{19}(C_{A2} + \max\{C_{A3}, C_{A3'}\})\alpha_{t-\tau_{\alpha_t}, t-1}(\|w_t\| + 1) \quad \text{(Lemma 19)}$$
$$\leq C_{19}(C_{A2} + \max\{C_{A3}, C_{A3'}\})\alpha_{t-\tau_{\alpha_t}, t-1}(\|w_t - w_{\text{ref}}\| + C_{\text{ref}} + 1).$$

Combining the two inequalities together yields

$$\langle \nabla L(w_{t-\tau_{\alpha_t}}), H(w_t, Y_{t+1}) - H(w_{t-\tau_{\alpha_t}}, Y_{t+1}) + h(w_{t-\tau_{\alpha_t}}) - h(w_t)\rangle$$
$$\leq L(2\|w_t - w_{\text{ref}}\| + 1) \cdot C_{19}(C_{A2} + \max\{C_{A3}, C_{A3'}\})\alpha_{t-\tau_{\alpha_t}, t-1}(\|w_t - w_{\text{ref}}\| + C_{\text{ref}} + 1)$$
$$\leq C_{2,1}\alpha_{t-\tau_{\alpha_t}, t-1}(\|w_t - w_{\text{ref}}\| + 1)^2.$$

Choosing $C_2 \doteq 4C_{2,1}$ then completes the proof. $\qquad\square$

## B.3. Proof of Lemma 3

*Proof.* For the first component of $T_3$ in (10), we have

$$\mathbb{E}\big[T_{31}\big|\mathcal{F}_{t-\tau_{\alpha_t}}\big] = \mathbb{E}\Big[\big\langle \nabla L(w_{t-\tau_{\alpha_t}}), H(w_{t-\tau_{\alpha_t}}, \tilde{Y}_{t+1}) - h(w_{t-\tau_{\alpha_t}})\big\rangle \big|\mathcal{F}_{t-\tau_{\alpha_t}}\Big]$$
$$\leq \Big\langle \nabla L(w_{t-\tau_{\alpha_t}}), \mathbb{E}\Big[H(w_{t-\tau_{\alpha_t}}, \tilde{Y}_{t+1}) - h(w_{t-\tau_{\alpha_t}})\big|\mathcal{F}_{t-\tau_{\alpha_t}}\Big]\Big\rangle$$
$$\leq \big\|\nabla L(w_{t-\tau_{\alpha_t}})\big\|_* \cdot \Big\|\mathbb{E}\Big[H(w_{t-\tau_{\alpha_t}}, \tilde{Y}_{t+1}) - h(w_{t-\tau_{\alpha_t}})\big|\mathcal{F}_{t-\tau_{\alpha_t}}\Big]\Big\|.$$

The first term is bounded in (21). For the second term, we have

$$\Big\|\mathbb{E}\Big[H(w_{t-\tau_{\alpha_t}}, \tilde{Y}_{t+1}) - h(w_{t-\tau_{\alpha_t}})\big|\mathcal{F}_{t-\tau_{\alpha_t}}\Big]\Big\|$$
$$= \Big\|\sum_y H(w_{t-\tau_{\alpha_t}}, y)P(\tilde{Y}_{t+1} = y|\mathcal{F}_{t-\tau_{\alpha_t}}) - \sum_y H(w_{t-\tau_{\alpha_t}}, y)d_{\mathcal{Y}, w_{t-\tau_{\alpha_t}}}(y)\Big\|$$
$$= \Big\|\sum_y H(w_{t-\tau_{\alpha_t}}, y)(P(\tilde{Y}_{t+1} = y|\mathcal{F}_{t-\tau_{\alpha_t}}) - d_{\mathcal{Y}, w_{t-\tau_{\alpha_t}}}(y))\Big\|$$
$$\leq \sum_y \big\|H(w_{t-\tau_{\alpha_t}}, y)\big\| \cdot \Big|P(\tilde{Y}_{t+1} = y|\mathcal{F}_{t-\tau_{\alpha_t}}) - d_{\mathcal{Y}, w_{t-\tau_{\alpha_t}}}(y)\Big|$$
$$\leq \max_y \big\|H(w_{t-\tau_{\alpha_t}}, y)\big\| \sum_y \Big|P(\tilde{Y}_{t+1} = y|\mathcal{F}_{t-\tau_{\alpha_t}}) - d_{\mathcal{Y}, w_{t-\tau_{\alpha_t}}}(y)\Big|$$
$$\leq \alpha_t C_{18}(\big\|w_{t-\tau_{\alpha_t}}\big\| + 1) \quad \text{(By (7), (8) and Lemma 18)}$$
$$\leq \alpha_t C_{18}(\big\|w_{t-\tau_{\alpha_t}} - w_t\big\| + \|w_t - w_{\text{ref}}\| + C_{\text{ref}} + 1).$$

Combining the two bounds yields

$$\mathbb{E}\Big[\langle \nabla L(w_{t-\tau_{\alpha_t}}), H(w_{t-\tau_{\alpha_t}}, \tilde{Y}_{t+1}) - h(w_{t-\tau_{\alpha_t}})\rangle |\mathcal{F}_{t-\tau_{\alpha_t}}\Big]$$
$$\leq LC_{18}\alpha_t(2\|w_t - w_{\text{ref}}\| + C_{\text{ref}} + C_{19})(C_{19}\alpha_{t-\tau_{\alpha_t}, t-1}(\|w_t\| + 1) + \|w_t - w_{\text{ref}}\| + C_{\text{ref}} + 1) \quad \text{(Lemma 19)}$$
$$\leq LC_{18}C_{19}\alpha_t(\|w_t - w_{\text{ref}}\| + C_{\text{ref}} + C_{19} + C_{3,1})^2$$
$$\leq LC_{18}C_{19}\alpha_t(\|w_t - w_{\text{ref}}\|^2 + C_{3,2}).$$

Choosing $C_3 \doteq LC_{18}C_{19}C_{3,2}$ then completes the proof. $\qquad\square$

## B.4. Proof of Lemma 4

We start with an auxiliary result.

**Lemma 20.** *Then there exist a constant $C_{20}$ such that*

$$\mathbb{E}\left[\left\|H(w_{t-\tau_{\alpha_t}}, Y_{t+1}) - H(w_{t-\tau_{\alpha_t}}, \tilde{Y}_{t+1})\right\|\Big|\mathcal{F}_{t-\tau_{\alpha_t}}\right] \le C_{20}\alpha_{t-\tau_{\alpha_t},t-1}\ln(t+t_0+1)(\|w_t - w_{ref}\| + 1).$$

*Proof.* In this proof, all $\Pr$ are probabilities implicitly conditioned on $\mathcal{F}_{t-\tau_{\alpha_t}}$. For simplicity, we define $P_t \doteq P_{w_t}$.

$$\Pr(Y_t = y') = \sum_y \Pr(Y_t = y', Y_{t-1} = y) = \sum_y P_t(y, y')\Pr(Y_{t-1} = y),$$

$$\Pr\left(\tilde{Y}_t = y'\right) = \sum_y \Pr\left(\tilde{Y}_{t-1} = y\right)P_{t-\tau_{\alpha_t}}(y, y').$$

Consequently,

$$\sum_{y'}\left|\Pr(Y_t = y') - \Pr\left(\tilde{Y}_t = y'\right)\right| \le \sum_{y,y'}\left|\Pr(Y_{t-1} = y)P_t(y, y') - \Pr\left(\tilde{Y}_{t-1} = y\right)P_{t-\tau_{\alpha_t}}(y, y')\right|.$$

We now consider two cases.

### Case 1. Under Assumption A3

$$\left|\Pr(Y_{t-1} = y)P_t(y, y') - \Pr\left(\tilde{Y}_{t-1} = y\right)P_{t-\tau_{\alpha_t}}(y, y')\right|$$

$$\le\left|\Pr(Y_{t-1} = y)P_t(y, y') - P(\tilde{Y}_{t-1} = y)P_t(y, y')\right| + \left|P(\tilde{Y}_{t-1} = y)P_t(y, y') - P(\tilde{Y}_{t-1} = y)P_{t-\tau_{\alpha_t}}(y, y')\right|$$

$$\le\left|\Pr(Y_{t-1} = y) - \Pr\left(\tilde{Y}_{t-1} = y\right)\right|P_t(y, y') + C_{A3}\frac{\|w_t - w_{t-\tau_{\alpha_t}}\|}{\|w_t\| + \|w_{t-\tau_{\alpha_t}}\| + 1}\Pr\left(\tilde{Y}_{t-1} = y\right)$$

$$\le\left|\Pr(Y_{t-1} = y) - \Pr\left(\tilde{Y}_{t-1} = y\right)\right|P_t(y, y') + C_{A3}C_{19}\frac{\alpha_{t-\tau_{\alpha_t},t-1}(\|w_t\| + 1)}{\|w_t\| + \|w_{t-\tau_{\alpha_t}}\| + 1}\Pr\left(\tilde{Y}_{t-1} = y\right) \quad \text{(Lemma 19)}$$

$$\le\left|\Pr(Y_{t-1} = y) - \Pr\left(\tilde{Y}_{t-1} = y\right)\right|P_t(y, y') + C_{20,1}\alpha_{t-\tau_{\alpha_t},t-1},$$

where $C_{20,1} \doteq C_{A3}C_{19}$.

### Case 2. Under Assumption A3'

$$\left|\Pr(Y_{t-1} = y)P_t(y, y') - \Pr\left(\tilde{Y}_{t-1} = y\right)P_{t-\tau_{\alpha_t}}(y, y')\right|$$

$$\le\left|\Pr(Y_{t-1} = y)P_t(y, y') - \Pr\left(\tilde{Y}_{t-1} = y\right)P_t(y, y')\right| + \left|\Pr\left(\tilde{Y}_{t-1} = y\right)P_t(y, y') - \Pr\left(\tilde{Y}_{t-1} = y\right)P_{t-\tau_{\alpha_t}}(y, y')\right|$$

$$\le\left|\Pr(Y_{t-1} = y) - \Pr\left(\tilde{Y}_{t-1} = y\right)\right|P_t(y, y') + C_{A3'}\|w_t - w_{t-\tau_{\alpha_t}}\|\Pr\left(\tilde{Y}_{t-1} = y\right)$$

$$\le\left|\Pr(Y_{t-1} = y) - \Pr\left(\tilde{Y}_{t-1} = y\right)\right|P_t(y, y') + C_{A3'}C_{19}\alpha_{t-\tau_{\alpha_t},t-1}(\|w_t\| + 1)\Pr\left(\tilde{Y}_{t-1} = y\right) \quad \text{(Lemma 19)}$$

$$\le\left|\Pr(Y_{t-1} = y) - \Pr\left(\tilde{Y}_{t-1} = y\right)\right|P_t(y, y') + C_{A3'}C_{19}\alpha_{t-\tau_{\alpha_t},t-1}(U_{A3'} + 1)\Pr\left(\tilde{Y}_{t-1} = y\right)$$

$$\le\left|\Pr(Y_{t-1} = y) - \Pr\left(\tilde{Y}_{t-1} = y\right)\right|P_t(y, y') + C_{20,2}\alpha_{t-\tau_{\alpha_t},t-1},$$

where $C_{20,2} \doteq C_{A3'}C_{19}(U_{A3'} + 1)$.

Thus, denote $C_{20,3} \doteq \max\{C_{20,1}, C_{20,2}\}$, we have

$$\sum_{y'}\left|\Pr(Y_t = y') - \Pr\left(\tilde{Y}_t = y'\right)\right| \le \sum_y\left|\Pr(Y_{t-1} = y) - \Pr\left(\tilde{Y}_{t-1} = y\right)\right| + C_{20,3}|\mathcal{Y}|\alpha_{t-\tau_{\alpha_t},t-1}.$$

Applying the above inequality recursively yields

$$\sum_{y'} \left| \Pr(Y_t = y') - \Pr\left(\tilde{Y}_t = y'\right) \right| \leq C_{20,3} |\mathcal{Y}| \sum_{j=t-\tau_{\alpha_t}}^{t-1} \alpha_{t-\tau_{\alpha_t},j}$$

For the summation term, we have

$$\frac{\sum_{j=t-\tau_{\alpha_t}}^{t-1} \alpha_{t-\tau_{\alpha_t},j}}{\alpha_{t-\tau_{\alpha_t},t-1}} \leq \frac{\tau_{\alpha_t} \tau_{\alpha_t} \alpha_{t-\tau_{\alpha_t}}}{\tau_{\alpha_t} \alpha_t} = \frac{\tau_{\alpha_t} \alpha_{t-\tau_{\alpha_t}}}{\alpha_t} = \mathcal{O}\left(\frac{\ln(t+t_0) \cdot \alpha_{t-\tau_{\alpha_t}}}{\alpha_t}\right) = \mathcal{O}\left(\ln(t+t_0)\right).$$

Then there exists a constant $C_{20,4}$ such that

$$\sum_{j=t-\tau_{\alpha_t}}^{t-1} \alpha_{t-\tau_{\alpha_t},j} \leq \alpha_{t-\tau_{\alpha_t},t-1} C_{20,4} \ln(t+t_0),$$

Consequently,

$$\mathbb{E}\left[ \left\| H(w_{t-\tau_{\alpha_t}}, Y_{t+1}) - H(w_{t-\tau_{\alpha_t}}, \tilde{Y}_{t+1}) \right\| \Big| \mathcal{F}_{t-\tau_{\alpha_t}} \right]$$

$$= \left\| \sum_{y'} H(w_{t-\tau_{\alpha_t}}, y') \Pr(Y_{t+1} = y') - \sum_{y'} H(w_{t-\tau_{\alpha_t}}, y') \Pr\left(\tilde{Y}_{t+1} = y'\right) \right\|$$

$$= \left\| \sum_{y'} H(w_{t-\tau_{\alpha_t}}, y') \Pr(Y_{t+1} = y') - H(w_{t-\tau_{\alpha_t}}, y') \Pr\left(\tilde{Y}_{t+1} = y'\right) \right\|$$

$$\leq C_{18}(\| w_{t-\tau_{\alpha_t}} \| + 1) \cdot \sum_{y'} \left| \Pr(Y_{t+1} = y') - \Pr\left(\tilde{Y}_{t+1} = y'\right) \right|$$

$$\leq C_{18} \alpha_{t-\tau_{\alpha_t},t-1}(C_{19}(\| w_t - w_{\text{ref}} \| + 1) + 1) C_{20,3} |\mathcal{Y}| C_{20,4} \ln(t+t_0+1).$$

Choosing $C_{20} \doteq C_{18}(C_{19}+1)C_{20,3}C_{20,4}|\mathcal{Y}|$ then completes the proof. $\square$

We are now ready to present the proof of Lemma 4.

*Proof.* For the second component of $T_3$ in (10), we have

$$\mathbb{E}\left[T_{32} | \mathcal{F}_{t-\tau_{\alpha_t}}\right] = \mathbb{E}\left[ \left\langle \nabla L(w_{t-\tau_{\alpha_t}}), H(w_{t-\tau_{\alpha_t}}, Y_{t+1}) - H(w_{t-\tau_{\alpha_t}}, \tilde{Y}_{t+1}) \right\rangle \Big| \mathcal{F}_{t-\tau_{\alpha_t}} \right]$$

$$\leq \left\langle \nabla L(w_{t-\tau_{\alpha_t}}), \mathbb{E}\left[ H(w_{t-\tau_{\alpha_t}}, Y_{t+1}) - H(w_{t-\tau_{\alpha_t}}, \tilde{Y}_{t+1}) \big| \mathcal{F}_{t-\tau_{\alpha_t}} \right] \right\rangle$$

$$\leq \left\| \nabla L(w_{t-\tau_{\alpha_t}}) \right\|_* \cdot \left\| \mathbb{E}\left[ H(w_{t-\tau_{\alpha_t}}, Y_{t+1}) - H(w_{t-\tau_{\alpha_t}}, \tilde{Y}_{t+1}) \big| \mathcal{F}_{t-\tau_{\alpha_t}} \right] \right\|.$$

From Lemma 20, we have

$$\mathbb{E}\left[ \left\| H(w_{t-\tau_{\alpha_t}}, Y_{t+1}) - H(w_{t-\tau_{\alpha_t}}, \tilde{Y}_{t+1}) \right\| \Big| \mathcal{F}_{t-\tau_{\alpha_t}} \right] \leq C_{20} \alpha_{t-\tau_{\alpha_t},t-1} \ln(t+t_0+1)(\| w_t - w_{\text{ref}} \| + 1).$$

Thus, there exists a constant $C_4$ such that

$$T_{32} \leq \mathbb{E}\left[ \left\| \nabla L(w_{t-\tau_{\alpha_t}}) \right\|_* \left\| \mathbb{E}\left[ H(w_{t-\tau_{\alpha_t}}, Y_{t+1}) - H(w_{t-\tau_{\alpha_t}}, \tilde{Y}_{t+1}) \big| \mathcal{F}_{t-\tau_{\alpha_t}} \right] \right\| \right]$$

$$\leq L(2\| w_t - w_{\text{ref}} \| + C_{\text{ref}} + C_{19}) C_{20} \alpha_{t-\tau_{\alpha_t},t-1} \ln(t+t_0+1)(\| w_t - w_{\text{ref}} \| + 1) \quad \text{(By (21))}$$

$$\leq 2LC_{20} \alpha_{t-\tau_{\alpha_t},t-1} \ln(t+t_0+1)(\| w_t - w_{\text{ref}} \| + C_{\text{ref}} + C_{19})^2$$

$$\leq C_4 \alpha_{t-\tau_{\alpha_t},t-1} \ln(t+t_0+1)(\| w_t - w_{\text{ref}} \|^2 + 1).$$

This completes the proof. $\square$

## B.5. Proof of Theorem 3

Recall the decomposition (9), combining the bounds for $T_1$, $T_2$, $T_{31}$ and $T_{32}$ from the above lemmas, we have:

$$\mathbb{E}\big[\langle \nabla L(w_t), H(w_t, Y_{t+1}) - h(w_t)\rangle | \mathcal{F}_{t-\tau_{\alpha_t}}\big]$$
$$\leq C_1 \alpha_{t-\tau_{\alpha_t}, t-1}(\mathbb{E}\big[L(w_t)|\mathcal{F}_{t-\tau_{\alpha_t}}\big] + 1) + C_2 \alpha_{t-\tau_{\alpha_t}, t-1}(\mathbb{E}\big[L(w_t)|\mathcal{F}_{t-\tau_{\alpha_t}}\big] + 1) + C_3 \alpha_t(\mathbb{E}\big[L(w_t)|\mathcal{F}_{t-\tau_{\alpha_t}}\big] + 1)$$
$$\quad + C_4 \alpha_{t-\tau_{\alpha_t}, t-1} \ln(t + t_0 + 1)(\mathbb{E}\big[L(w_t)|\mathcal{F}_{t-\tau_{\alpha_t}}\big] + 1)$$
$$\leq D_3 \alpha_{t-\tau_{\alpha_t}, t-1} \ln(t + t_0 + 1)(\mathbb{E}\big[L(w_t)|\mathcal{F}_{t-\tau_{\alpha_t}}\big] + 1).$$

where $D_3 \doteq C_1 + C_2 + C_3 + C_4$ is constant. Recall (6) and combine all the results, we have

$$\mathbb{E}\big[L(w_{t+1})|\mathcal{F}_{t-\tau_{\alpha_t}}\big]$$
$$\leq \mathbb{E}\big[L(w_t)|\mathcal{F}_{t-\tau_{\alpha_t}}\big] + \alpha_t \mathbb{E}\big[\langle \nabla L(w_t), h(w_t)\rangle | \mathcal{F}_{t-\tau_{\alpha_t}}\big]$$
$$\quad + \alpha_t \mathbb{E}\big[\langle \nabla L(w_t), H(w_t, Y_{t+1}) - h(w_t)\rangle | \mathcal{F}_{t-\tau_{\alpha_t}}\big] + \frac{L\alpha_t^2}{2}\mathbb{E}\big[\|H(w_t, Y_{t+1})\|^2 | \mathcal{F}_{t-\tau_{\alpha_t}}\big]$$
$$\leq \mathbb{E}\big[L(w_t)|\mathcal{F}_{t-\tau_{\alpha_t}}\big] + \alpha_t \mathbb{E}\big[\langle \nabla L(w_t), h(w_t)\rangle | \mathcal{F}_{t-\tau_{\alpha_t}}\big]$$
$$\quad + \alpha_t D_3 \alpha_{t-\tau_{\alpha_t}, t-1} \ln(t + t_0 + 1)(\mathbb{E}\big[L(w_t)|\mathcal{F}_{t-\tau_{\alpha_t}}\big] + 1) + \frac{L\alpha_t^2}{2}C_{18}^2(1 + \|w_t - w_{\text{ref}}\|)^2$$
$$\leq \mathbb{E}\big[L(w_t)|\mathcal{F}_{t-\tau_{\alpha_t}}\big] + \alpha_t \mathbb{E}\big[\langle \nabla L(w_t), h(w_t)\rangle | \mathcal{F}_{t-\tau_{\alpha_t}}\big]$$
$$\quad + \alpha_t D_3 \alpha_{t-\tau_{\alpha_t}, t-1} \ln(t + t_0 + 1)(\mathbb{E}\big[L(w_t)|\mathcal{F}_{t-\tau_{\alpha_t}}\big] + 1) + 2L\alpha_t^2 C_{18}^2(\mathbb{E}\big[L(w_t)|\mathcal{F}_{t-\tau_{\alpha_t}}\big] + 1).$$

Denoting $f(t) \doteq D_3 \alpha_t \alpha_{t-\tau_{\alpha_t}, t-1} \ln(t + t_0 + 1) + \frac{2L\alpha_t^2 C_{A2}^2}{l_s^2} = \mathcal{O}\left(\frac{\ln^2(t+t_0+1)}{(t+t_0)^{2\epsilon_\alpha}}\right)$ and taking the total expectation then completes the proof of Theorem 3.

# C. Proofs in Section 5.2

## C.1. Proof of Lemma 5

*Proof.* Since softmax is Lipschitz continuous, we only need to bound $\big|\kappa_{w_1} x(s,a)^\top w_1 - \kappa_{w_2} x(s,a)^\top w_2\big|$.

**Case 1:** $\|w_1\|_2 < 1$ **and** $\|w_2\|_2 < 1$. In this case, $\kappa_{w_1} = \kappa_{w_2} = \kappa_0$. Define $C_x \doteq \sup_{s,a} \|x(s,a)\|_2$. Then we have

$$\big|\kappa_{w_1} x(s,a)^\top w_1 - \kappa_{w_2} x(s,a)^\top w_2\big|$$
$$\leq \kappa_0 C_x \|w_1 - w_2\|_2$$
$$\leq \frac{3\kappa_0 C_x}{1 + \|w_1\|_2 + \|w_2\|_2}\|w_1 - w_2\|_2.$$

**Case 2:** $\|w_1\|_2 \geq 1$ **and** $\|w_2\|_2 \geq 1$. Without loss of generality, let $\|w_1\| \geq \|w_2\|$. In this case, $\kappa_{w_1} = \frac{\kappa_0}{\|w_1\|_2}, \kappa_{w_2} = \frac{\kappa_0}{\|w_2\|_2}$. Then we have

$$|\kappa_{w_1} - \kappa_{w_2}| = \kappa_0 \left|\frac{\|w_2\|_2 - \|w_1\|_2}{\|w_1\|_2 \|w_2\|_2}\right| \leq \kappa_0 \frac{\|w_1 - w_2\|_2}{\|w_1\|_2 \|w_2\|_2}.$$

Therefore,

$$\big|\kappa_{w_1} x(s,a)^\top w_1 - \kappa_{w_2} x(s,a)^\top w_2\big|$$
$$\leq \big\|\kappa_{w_1} x(s,a)^\top (w_1 - w_2)\big\|_2 + \big|x(s,a)^\top w_2\big||\kappa_{w_1} - \kappa_{w_2}|$$
$$\leq \frac{\kappa_0}{\|w_1\|_2}C_x \|w_1 - w_2\|_2 + C_x \|w_2\|_2 \kappa_0 \frac{\|w_1 - w_2\|_2}{\|w_1\|_2 \|w_2\|_2}$$
$$\leq \frac{6\kappa_0 C_x}{3\|w_1\|_2}\|w_1 - w_2\|_2$$
$$\leq \frac{6\kappa_0 C_x}{1 + \|w_1\|_2 + \|w_2\|_2}\|w_1 - w_2\|_2.$$

**Case 3: $\|w_1\|_2 < 1$ and $\|w_2\|_2 \geq 1$, and vice versa.** In this case, $\kappa_{w_1} = \kappa_0$, and $\kappa_{w_2} = \frac{\kappa_0}{\|w_2\|_2} \leq \kappa_0$. We can obtain that

$$|\kappa_{w_1} - \kappa_{w_2}| = \kappa_0\left(1 - \frac{1}{\|w_2\|_2}\right).$$

Similarly, we have

$$
\begin{aligned}
&\left|\kappa_{w_1} x(s,a)^\top w_1 - \kappa_{w_2} x(s,a)^\top w_2\right| \\
\leq& \left\|\kappa_{w_2} x(s,a)^\top (w_1 - w_2)\right\|_2 + \left|x(s,a)^\top w_1\right|\left|\kappa_{w_1} - \kappa_{w_2}\right| \\
\leq& \frac{\kappa_0 C_x}{\|w_2\|}\|w_1 - w_2\|_2 + C_x\|w_1\|_2\kappa_0\left(1 - \frac{1}{\|w_2\|_2}\right) \\
\leq& \frac{\kappa_0 C_x}{\|w_2\|}\|w_1 - w_2\|_2 + C_x\kappa_0\left(1 - \frac{1}{\|w_2\|_2}\right) \\
\leq& \frac{\kappa_0 C_x}{\|w_2\|}\|w_1 - w_2\|_2 + \frac{C_x\kappa_0}{\|w_2\|_2}(\|w_2\|_2 - 1) \\
\overset{(*)}{\leq}& \frac{6\kappa_0 C_x}{3\|w_2\|_2}\|w_1 - w_2\|_2 \\
\leq& \frac{6\kappa_0 C_x}{1 + \|w_1\|_2 + \|w_2\|_2}\|w_1 - w_2\|_2,
\end{aligned}
$$

where $(*)$ is obtained because $\|w_1\|_2 < 1$, and according to the triangle inequality $\|w_1 - w_2\|_2 \geq \|w_2\|_2 - \|w_1\|_2 \geq \|w_2\|_2 - 1$ since $\|w_2\|_2 \geq 1$. This completes the proof. $\qquad\square$

### C.2. Proof of Lemma 6

*Proof.* According to the definition in (13) and (14), we can apply the triangle inequality to get

$$
\begin{aligned}
&\|h(w_1) - h(w_2)\|_2 \\
=&\|A(w_1)w_1 + b(w_1) - A(w_2)w_2 - b(w_2)\|_2 \\
\leq&\left\|X^\top D_{\mu_{w_1}}(\mathcal{T}(Xw_1) - Xw_1) - X^\top D_{\mu_{w_2}}(\mathcal{T}(Xw_2) - Xw_2)\right\|_2 \\
\leq&\left\|X^\top D_{\mu_{w_1}}(\mathcal{T}(Xw_1) - \mathcal{T}(Xw_2) - (Xw_1 - Xw_2))\right\|_2 + \left\|X^\top(D_{\mu_{w_1}} - D_{\mu_{w_2}})(\mathcal{T}(Xw_2) - Xw_2)\right\|_2.
\end{aligned}
$$

The first term in the RHS can be bounded by $\|w_1 - w_2\|$ easily because $\left\|D_{\mu_{w_1}}\right\|_2 \leq 1$ and $\mathcal{T}$ is a contraction (and thus Lipschitz continuous w.r.t. any norm). For the second term in the RHS, according to Lemma 15, $D_{\mu_w}$ is Lipschitz continuous on $\mu_w$, that is

$$\left\|D_{\mu_{w_1}} - D_{\mu_{w_2}}\right\|_2 \leq C_{6,1}\|\mu_{w_1} - \mu_{w_2}\|_2.$$

Here we interpret $\mu_w$ as a vector in $\mathbb{R}^{|\mathcal{S}||\mathcal{A}|}$. It is easy to see that

$$\|\mathcal{T}(Xw_2) - Xw_2\| \leq C_{6,2} + C_{6,2}\|w_2\|_2$$

for some $C_{6,2}$. Then Lemma 5 implies that for some $C_{6,3}$, we have

$$
\begin{aligned}
&\left\|X^\top(D_{\mu_{w_1}} - D_{\mu_{w_1}})(\mathcal{T}(Xw_2) - Xw_2)\right\|_2 \\
\leq&\|X\|_2 C_{6,1}\|\mu_{w_1} - \mu_{w_2}\|_2(C_{6,2} + C_{6,2}\|w_2\|_2) \\
\leq& C_{6,3}\frac{1 + \|w_2\|_2}{1 + \|w_1\|_2 + \|w_2\|_2}\|w_1 - w_2\|_2 \\
\leq& C_{6,3}\|w_1 - w_2\|_2,
\end{aligned}
$$

which completes the proof. $\qquad\square$

### C.3. Proof of Lemma 7

*Proof.* From Lemma 16, we know that if $\|w\|_2 \geq 1$, there exist positive constants $\beta$, which satisfy

$$w^\top A(w)w \leq -\beta\|w\|_2^2.$$

Therefore, for $\|w\|_2 \geq 1$, recall $b(w) = X^\top D_{\mu_w}r$, we have a constant $C_{7,1} \doteq |\mathcal{S}|^2|\mathcal{A}|C_xC_r$ that ensures

$$\langle w, A(w)w + b(w)\rangle \leq -\beta\|w\|_2^2 + C_{7,1}\|w\|_2.$$

For $\|w\|_2 \leq 1$, recall $\|A\|_2$ can be bounded by another constant $C_{7,2} = (|\mathcal{S}||\mathcal{A}|C_x)^2(\gamma+1)$, thus

$$\langle w, A(w)w + b(w)\rangle \leq (C_{7,1} + C_{7,2})\|w\|_2 \leq -\beta\|w\|_2^2 + (C_{7,1} + C_{7,2} + \beta)\|w\|_2.$$

Thus, for $C_7 \doteq C_{7,1} + C_{7,2} + \beta = |\mathcal{S}|^2|\mathcal{A}|C_xC_r + (|\mathcal{S}||\mathcal{A}|C_x)^2(\gamma+1) + \beta$, it always holds that

$$\langle w_t, A(w_t)w_t + b(w_t)\rangle \leq -\beta\|w_t\|_2^2 + C_7\|w_t\|_2.$$

This completes the proof. $\qquad\square$

### C.4. Proof of Theorem 1

*Proof.* Combining (15) and Lemma 7, we have

$$
\begin{aligned}
&\mathbb{E}[L(w_{t+1})]\\
\leq& \mathbb{E}[L(w_t)] + \alpha_t\mathbb{E}[\langle \nabla L(w_t), h(w_t)\rangle] + f(t)(1 + \mathbb{E}[L(w_t)])\\
\leq& \mathbb{E}[L(w_t)] + \alpha_t(-\beta\mathbb{E}\big[\|w_t\|_2^2\big] + C_7\mathbb{E}[\|w_t\|_2]) + f(t)(1 + \mathbb{E}[L(w_t)])\\
=& (1 - 2\beta\alpha_t + f(t))\mathbb{E}[L(w_t)] + \alpha_t C_7\mathbb{E}[\|w_t\|_2] + f(t)\\
\leq& (1 - 2\beta\alpha_t + f(t))\mathbb{E}[L(w_t)] + \mathbb{E}\left[\alpha_t\beta\cdot\frac{1}{2}\|w_t\|_2^2 + \frac{2\alpha_t}{\beta}C_7^2\right] + f(t) \quad \text{(using } x + y \geq \sqrt{xy})\\
\leq& (1 - \beta\alpha_t + f(t))\mathbb{E}[L(w_t)] + D_{1,1}\alpha_t \quad \text{(since } f(t) \leq \frac{2C_7^2}{\beta}\alpha_t \text{ for sufficiently large } t)\\
\leq& (1 - D_{1,2}\alpha_t)\mathbb{E}[L(w_t)] + D_{1,1}\alpha_t.
\end{aligned}
$$

where $D_{1,1} \doteq \frac{4}{\beta}C_7^2$ and $D_{1,2} \doteq \frac{\beta}{2}$. Unfolding the recursion from time $\bar{t}$ to $t$ yields

$$\mathbb{E}[L(w_t)] \leq \prod_{k=\bar{t}}^{t-1}(1 - D_{1,2}\alpha_k)\mathbb{E}[L(w_{\bar{t}})] + D_{1,1}\sum_{k=\bar{t}}^{t-1}\alpha_k\prod_{j=k+1}^{t-1}(1 - D_{1,2}\alpha_j).$$

**Case 1:** $\epsilon_\alpha = 1$. We can derive that

$$
\begin{aligned}
\prod_{j=k+1}^{t-1}(1 - D_{1,2}\alpha_j) &\leq \exp\left(-D_{1,2}\sum_{j=k+1}^{t-1}\alpha_j\right)\\
&\leq \exp\left(-D_{1,2}\alpha\int_{k+1}^t\frac{1}{x+t_0}dx\right)\\
&= \exp\left(-D_{1,2}\alpha(\ln(t+t_0) - \ln(k+1+t_0))\right)\\
&= \left(\frac{k+1+t_0}{t+t_0}\right)^{D_{1,2}\alpha}.
\end{aligned}
$$

Substituting back, we obtain

$$\mathbb{E}[L(w_t)] \leq \left(\frac{t_0+\bar{t}}{t+t_0}\right)^{D_{1,2}\alpha}\mathbb{E}[L(w_{\bar{t}})] + D_{1,1}\sum_{k=\bar{t}}^{t-1}\frac{\alpha}{k+t_0}\left(\frac{k+1+t_0}{t+t_0}\right)^{D_{1,2}\alpha}.$$

The second term becomes

$$\sum_{k=\bar{t}}^{t-1} \frac{\alpha}{k+t_0} \left(\frac{k+1+t_0}{t+t_0}\right)^{D_{1,2}\alpha} \leq \frac{2^{D_{1,2}}\alpha}{(t+t_0)^{D_{1,2}\alpha}} \sum_{k=\bar{t}}^{t-1} (k+t_0)^{D_{1,2}\alpha-1}.$$

Denote $c \doteq D_{1,2}\alpha$ and $S(t) \doteq \frac{1}{(t+t_0)^c} \sum_{k=\bar{t}}^{t-1}(k+t_0)^{c-1}$.

1. When $c \neq 1$:

$$\sum_{k=\bar{t}}^{t-1}(k+t_0)^{c-1} \leq \int_{\bar{t}-1}^{t}(x+t_0)^{c-1}dx \leq \frac{(t+t_0)^c}{c},$$

$$S(t) \leq \frac{1}{(t+t_0)^c} \cdot \frac{(t+t_0)^c}{c} = \frac{1}{c}.$$

2. When $c = 1$:

$$S(t) \leq \frac{1}{t+t_0} \sum_{k=\bar{t}}^{t-1} k^0 = \frac{t-\bar{t}}{t+t_0} < 1.$$

Thus, we conclude that $S(t) \leq \frac{1}{D_{1,2}\alpha}$, that is

$$\mathbb{E}[L(w_t)] \leq \left(\frac{t_0+\bar{t}}{t+t_0}\right)^{D_{1,2}\alpha} \mathbb{E}[L(w_{\bar{t}})] + D_{1,1}2^{D_{1,2}}\frac{1}{D_{1,2}\alpha}$$

$$= \frac{D_{1,3}}{(t+t_0)^{D_{1,2}\alpha}}\mathbb{E}[L(w_{\bar{t}})] + D_{1,4}.$$

Furthermore, the last constant can be expanded as $D_{1,4} \leq \frac{4}{\beta}C_7^2\frac{2^\beta}{\beta/2} \leq ((|\mathcal{S}||\mathcal{A}|C_x)^2(\gamma+1) + |\mathcal{S}|^2|\mathcal{A}|C_xC_r)^2\frac{2^{\beta+3}}{\beta^2}$. For the $\mathbb{E}[L(w_{\bar{t}})]$ term, since $\bar{t}$ is deterministic, starting from the update of $w_{t+1}$, we have

$$\|w_{t+1}\| \leq \|w_t\| + \alpha_t\|H(w_t, Y_{t+1})\| \leq \|w_t\| + \alpha_t C_{18}(\|w_t\|+1).$$

That is, $\|w_{t+1}\| \leq \alpha_0 C_{18} + \sum_{i=0}^{t}(\alpha_0 C_{18}+1)\|w_i\|$. Applying discrete Gronwall inequality, we obtain

$$\|w_{\bar{t}}\| \leq (C_{18}+\|w_0\|)\exp\left(\sum_{t=0}^{\bar{t}-1}(1+\alpha_0 C_{18})\right) = (C_{18}+\|w_0\|)\exp(\bar{t}+\bar{t}\alpha_0 C_{18})$$

Furthermore, combining this with the current bound, denote $B_{1,1} \doteq D_{1,3}\exp(2\bar{t}(1+\alpha_0 C_{18}))$, $B_{1,2} \doteq D_{1,2}$ and $B_{1,3} \doteq 2\left(\frac{D_{1,3}}{(\bar{t}+t_0)^{D_{1,2}\alpha}} \times 2C_{18}\exp(2\bar{t}(1+\alpha_0 C_{18})) + D_{1,4}\right)$ then completes the proof of the first case.

**Case 2:** $\epsilon_\alpha \in (0,1)$.

$$\prod_{j=k+1}^{t-1}(1-D_{1,2}\alpha_j) \leq \exp\left(-D_{1,2}\sum_{j=k+1}^{t-1}\alpha_j\right)$$

$$\leq \exp\left(-D_{1,2}\alpha\int_{k+1}^{t}\frac{1}{(x+t_0)^{\epsilon_\alpha}}dx\right)$$

$$= \exp\left(-\frac{D_{1,2}\alpha}{1-\epsilon_\alpha}\left[(t+t_0)^{1-\epsilon_\alpha} - (k+1+t_0)^{1-\epsilon_\alpha}\right]\right)$$

$$= \frac{\exp\left(\frac{D_{1,2}\alpha}{1-\epsilon_\alpha}(k+1+t_0)^{1-\epsilon_\alpha}\right)}{\exp\left(\frac{D_{1,2}\alpha}{1-\epsilon_\alpha}(t+t_0)^{1-\epsilon_\alpha}\right)}.$$

Substituting back, we obtain

$$\mathbb{E}[L(w_t)] \leq \exp\left(-\frac{D_{1,2}\alpha}{1-\epsilon_\alpha}\left[(t+t_0)^{1-\epsilon_\alpha} - (\bar{t}+t_0)^{1-\epsilon_\alpha}\right]\right)\mathbb{E}[L(w_{\bar{t}})]$$
$$+ D_{1,1}\sum_{k=\bar{t}}^{t-1}\frac{\alpha}{(k+t_0)^{\epsilon_\alpha}}\exp\left(-\frac{D_{1,2}\alpha}{1-\epsilon_\alpha}\left[(t+t_0)^{1-\epsilon_\alpha} - (k+1+t_0)^{1-\epsilon_\alpha}\right]\right).$$

For the second term,

$$\sum_{k=\bar{t}}^{t-1}\frac{\alpha}{(k+t_0)^{\epsilon_\alpha}}\exp\left(-\frac{D_{1,2}\alpha}{1-\epsilon_\alpha}\left[(t+t_0)^{1-\epsilon_\alpha} - (k+1+t_0)^{1-\epsilon_\alpha}\right]\right)$$
$$\leq \frac{\alpha}{\exp\left(\frac{D_{1,2}\alpha}{1-\epsilon_\alpha}(t+t_0)^{1-\epsilon_\alpha}\right)}\sum_{k=\bar{t}}^{t-1}\frac{1}{(k+t_0)^{\epsilon_\alpha}}\exp\left(\frac{D_{1,2}\alpha}{1-\epsilon_\alpha}(k+1+t_0)^{1-\epsilon_\alpha}\right).$$

To make the notation cleaner, we define $M = \frac{D_{1,2}\alpha}{1-\epsilon_\alpha}$, then

$$RHS = \frac{\alpha}{\exp\left(M(t+t_0)^{1-\epsilon_\alpha}\right)}\sum_{k=\bar{t}+t_0}^{t+t_0-1}\frac{1}{k^{\epsilon_\alpha}}\exp\left(M(k+1)^{1-\epsilon_\alpha}\right)$$
$$\leq \frac{\alpha}{\exp\left(M(t+t_0)^{1-\epsilon_\alpha}\right)}\sum_{k=\bar{t}+t_0}^{t+t_0-1}\frac{2^{\epsilon_\alpha}}{(k+1)^{\epsilon_\alpha}}\exp\left(M(k+1)^{1-\epsilon_\alpha}\right)$$
$$\leq \frac{2^{\epsilon_\alpha}\alpha}{\exp\left(M(t+t_0)^{1-\epsilon_\alpha}\right)}\int_1^{t+t_0+1}x^{-\epsilon_\alpha}\exp\left(Mx^{1-\epsilon_\alpha}\right)dx.$$

Now we perform a substitution to simplify the integral, let $u \doteq Mx^{1-\epsilon_\alpha}$, then $du = M(1-\epsilon_\alpha)x^{-\epsilon_\alpha}dx$, thus

$$\int_1^{t+t_0+1}x^{-\epsilon_\alpha}\exp\left(Mx^{1-\epsilon_\alpha}\right)dx = \frac{1}{M(1-\epsilon_\alpha)}\int_M^{M(t+t_0+1)^{1-\epsilon_\alpha}}\exp\left(u\right)du \leq \frac{\exp\left(M(t+t_0+1)^{1-\epsilon_\alpha}\right)}{M(1-\epsilon_\alpha)}.$$

Finally, we have

$$\mathbb{E}[L(w_t)] \leq \exp\left(-\frac{D_{1,2}\alpha}{1-\epsilon_\alpha}\left[(t+t_0)^{1-\epsilon_\alpha} - (\bar{t}+t_0)^{1-\epsilon_\alpha}\right]\right)\mathbb{E}[L(w_{\bar{t}})]$$
$$+ D_{1,1}\sum_{k=\bar{t}}^{t-1}\frac{\alpha}{(k+t_0)^{\epsilon_\alpha}}\exp\left(-\frac{D_{1,2}\alpha}{1-\epsilon_\alpha}\left[(t+t_0)^{1-\epsilon_\alpha} - (k+1+t_0)^{1-\epsilon_\alpha}\right]\right)$$
$$\leq \exp\left(-\frac{D_{1,2}\alpha}{1-\epsilon_\alpha}\left[(t+t_0)^{1-\epsilon_\alpha} - (\bar{t}+t_0)^{1-\epsilon_\alpha}\right]\right)\mathbb{E}[L(w_{\bar{t}})]$$
$$+ D_{1,1}\frac{2^{\epsilon_\alpha}\alpha}{\exp\left(M(t+t_0)^{1-\epsilon_\alpha}\right)}\frac{\exp\left(M(t+t_0)^{1-\epsilon_\alpha}\right)}{M(1-\epsilon_\alpha)}$$
$$= \exp\left(-\frac{D_{1,2}\alpha}{1-\epsilon_\alpha}\left[(t+t_0)^{1-\epsilon_\alpha} - (\bar{t}+t_0)^{1-\epsilon_\alpha}\right]\right)\mathbb{E}[L(w_{\bar{t}})] + \frac{2^{\epsilon_\alpha}D_{1,1}\alpha}{M(1-\epsilon_\alpha)}$$
$$= \exp\left(-\frac{D_{1,2}\alpha}{1-\epsilon_\alpha}\left[(t+t_0)^{1-\epsilon_\alpha} - (\bar{t}+t_0)^{1-\epsilon_\alpha}\right]\right)\mathbb{E}[L(w_{\bar{t}})] + \frac{2^{\epsilon_\alpha}D_{1,1}}{D_{1,2}}$$
$$= \exp\left(-\frac{D_{1,2}\alpha}{1-\epsilon_\alpha}\left[(t+t_0)^{1-\epsilon_\alpha} - (\bar{t}+t_0)^{1-\epsilon_\alpha}\right]\right)\mathbb{E}[L(w_{\bar{t}})] + D_{1,5}.$$

The last constant can be expanded as $D_{1,5} = \frac{2^{\epsilon_\alpha+3}C_7^2}{\beta^2} = \frac{2^{\epsilon_\alpha+3}}{\beta^2}((|\mathcal{S}||\mathcal{A}|C_x)^2(\gamma+1) + |\mathcal{S}|^2|\mathcal{A}|C_xC_r)^2$. Then Theorem 1 follows from using the Gronwall inequality to bound $L(w_{\bar{t}})$ with $\|w_0\|$ and the equivalence of norms, with $B_{1,4}$, $B_{1,5}$, and $B_{1,6}$ selected similarly. $\qquad\square$

# D. Proofs in Section 5.3

## D.1. Proof of Lemma 8

*Proof.* It is obvious that $q_*$ is the unique fixed point because $\inf_{q,s,a} d_{\mu_q}(s,a) > 0$ thanks to the fact that $\epsilon > 0$. We also have

$$
\begin{aligned}
\left\|\mathcal{T}'q - q_*\right\|_\infty &= \left\|\mathcal{T}'q - D_{\mu_q}(q_* - q_*) - q_*\right\|_\infty \\
&= \left\|D_{\mu_q}\mathcal{T}q - D_{\mu_q}q + q - q_* + D_{\mu_q}(q_* - q_*)\right\|_\infty \\
&= \left\|D_{\mu_q}(\mathcal{T}q - q_*) + (I - D_{\mu_q})(q - q_*)\right\|_\infty \\
&\le \max_{s,a}\left|d_{\mu_q}(s,a)(\mathcal{T}q - q_*)(s,a) + (1 - d_{\mu_q}(s,a))(q - q_*)(s,a)\right| \\
&\le \max_{s,a} d_{\mu_q}(s,a)\|\mathcal{T}q - q_*\|_\infty + (1 - d_{\mu_q}(s,a))\|q - q_*\|_\infty \\
&\le \max_{s,a} d_{\mu_q}(s,a)\gamma\|q - q_*\|_\infty + (1 - d_{\mu_q}(s,a))\|q - q_*\|_\infty \\
&= \max_{s,a}(1 - (1 - \gamma)d_{\mu_q}(s,a))\|q - q_*\|_\infty \\
&\le \left(1 - (1 - \gamma)\inf_{q,s,a} d_{\mu_q}(s,a)\right)\|q - q_*\|_\infty,
\end{aligned}
$$

which completes the proof. $\qquad\square$

## D.2. Proof of Lemma 10

*Proof.* Recalling the update rule for tabular $Q$-learning in (SA) and (11), we have

$$
\begin{aligned}
\|q_{t+1}\|_\infty &\le (1 - \alpha_t)\|q_t\|_\infty + \alpha_t(C_r + \gamma\|q_t\|_\infty) \\
&= (1 - \alpha_t + \gamma\alpha_t)\|q_t\|_\infty + \alpha_t C_r.
\end{aligned}
$$

If $\|q_t\|_m$ is unbounded, then $\|q_t\|_\infty$ is also unbounded, that is for each constant $M > \|q_0\|_\infty$, we can find some time $t$ such that $\|q_t\|_\infty < M \le \|q_{t+1}\|_\infty$. Therefore,

$$
M < (1 - \alpha_t + \gamma\alpha_t)M + \alpha_t C_r,
$$

which is equivalent to

$$
\alpha_t(1 - \gamma)M < \alpha_t C_r.
$$

Since $C_r$ is a constant, we will get a contradiction for $M > \frac{C_r}{1-\gamma}$. This completes the proof. $\qquad\square$

## D.3. Proof of Lemma 11

*Proof.* Recalling the definition of $h(w)$ in (16), we have

$$
\begin{aligned}
\|h(q_1) - h(q_2)\|_\infty &= \|\mathcal{T}'q_1 - q_1 - (\mathcal{T}'q_2 - q_2)\|_\infty \\
&= \left\|D_{\mu_{q_1}}(\mathcal{T}q_1 - q_1) - D_{\mu_{q_1}}(\mathcal{T}q_2 - q_2) + D_{\mu_{q_1}}(\mathcal{T}q_2 - q_2) - D_{\mu_{q_2}}(\mathcal{T}q_2 - q_2)\right\|_\infty \\
&\le \left\|D_{\mu_{q_1}}(\mathcal{T}q_1 - q_1) - D_{\mu_{q_1}}(\mathcal{T}q_2 - q_2)\right\|_\infty + \left\|D_{\mu_{q_1}}(\mathcal{T}q_2 - q_2) - D_{\mu_{q_2}}(\mathcal{T}q_2 - q_2)\right\|_\infty \\
&\le \|(\mathcal{T}q_1 - \mathcal{T}q_2) - (q_1 - q_2)\|_\infty + \left\|D_{\mu_{q_1}} - D_{\mu_{q_2}}\right\|_\infty\|\mathcal{T}q_2 - q_2\|_\infty.
\end{aligned}
$$

The first term in the RHS can be bounded with $\|q_1 - q_2\|_\infty$ easily. For the second term in the RHS, $\left\|D_{\mu_{q_1}} - D_{\mu_{q_2}}\right\|$ can be bounded with $\|q_1 - q_2\|_\infty$ thanks to Lemma 15. Noticing that $\|\mathcal{T}q_2 - q_2\|_\infty$ is bounded because $q_2$ lies in a compact set then completes the proof. $\qquad\square$

## D.4. Proof of Lemma 12

*Proof.* Recall that $\beta_m = (1 - \gamma)\inf_{q,s,a} d_{\mu_q}(s,a)$, we have

$$
\langle \nabla M(q_t - q_*), h(q_t)\rangle
$$

$$
\begin{aligned}
&= \langle \nabla M(q_t - q_*), \mathcal{T}' q_t - q_t \rangle \\
&= \langle \nabla M(q_t - q_*), \mathcal{T}' q_t - q_* + q_* - q_t \rangle \\
&= \langle \nabla M(q_t - q_*), \mathcal{T}' q_t - q_* \rangle - \langle \nabla M(q_t - q_*), q_t - q_* \rangle \\
&\leq \|q_t - q_*\|_m \|\mathcal{T}' q_t - q_*\|_m - \|q_t - q_*\|_m^2 \quad \text{(Lemma 9)} \\
&\leq \|q_t - q_*\|_m \cdot \frac{1}{l_{im}} \|\mathcal{T}' q_t - q_*\|_\infty - \|q_t - q_*\|_m^2 \\
&\leq \|q_t - q_*\|_m \cdot \frac{1}{l_{im}} \beta_m \|q_t - q_*\|_\infty - \|q_t - q_*\|_m^2 \\
&= -\left(1 - \frac{u_{im}}{l_{im}} \beta_m \right) \|q_t - q_*\|_m^2.
\end{aligned}
$$

Since $\beta_m < 1$, we can choose a sufficiently small $\xi$ (defined in Lemma 9) such that $C_{12} \doteq 1 - \frac{u_{im}}{l_{im}} \beta_m > 0$, which completes the proof. $\qquad \square$

## D.5. Proof of Theorem 2

*Proof.* Combining (17) and Lemma 12 yields

$$
\begin{aligned}
\mathbb{E}[L(q_{t+1})] &\leq \mathbb{E}[L(q_t)] + \alpha_t \mathbb{E}[\langle \nabla L(q_t), h(q_t) \rangle] + f(t)(1 + \mathbb{E}[L(q_t)]) \\
&= (1 - 2C_{12}\alpha_t + f(t))\mathbb{E}[L(q_t)] + f(t) \\
&\leq (1 - D_{2,1}\alpha_t)\mathbb{E}[L(q_t)] + D_{2,2} \frac{\ln^2(t + t_0)}{(t + t_0)^{2\epsilon_\alpha}}.
\end{aligned}
$$

Recall we have:

$$
\alpha_t = \frac{\alpha}{(t + t_0)^{\epsilon_\alpha}}, \quad \alpha_{t-\tau, t-1} = \mathcal{O}\left( \frac{\ln(t + t_0)}{(t + t_0)^{\epsilon_\alpha}} \right),
$$

where $\epsilon_\alpha \in (0.5, 1]$ and $t_0 > 0$ is chosen sufficiently large. Then for any $0.5 < \epsilon_\alpha' < \epsilon_\alpha$, for $t$ large enough, the recursive inequality can be simplified to:

$$
\mathbb{E}[L(q_{t+1})] \leq (1 - D_{2,1}\alpha_t)\mathbb{E}[L(q_t)] + D_{2,2} \frac{\alpha^2}{(t + t_0)^{2\epsilon_\alpha'}}.
$$

Thus,

$$
\mathbb{E}[L(q_t)] \leq \left( \prod_{k=\bar{t}}^{t-1} (1 - D_{2,1}\alpha_k) \right) \mathbb{E}[L(q_{\bar{t}})] + D_{2,2} \sum_{k=\bar{t}}^{t-1} \frac{\alpha^2}{(t + t_0)^{2\epsilon_\alpha'}} \prod_{j=k+1}^{t-1} (1 - D_{2,1}\alpha_k).
$$

Therefore we have:

$$
\prod_{j=k+1}^{t-1} (1 - D_{2,1}\alpha_j) \leq \exp\left( -D_{2,1} \sum_{j=k+1}^{t-1} \alpha_j \right).
$$

**Case 1:** $\epsilon_\alpha = 1$. With the step size $\alpha_j = \frac{\alpha}{j + t_0}$, the sum can be approximated by an integral:

$$
\sum_{j=k+1}^{t-1} \alpha_j \geq \alpha \int_{k+1}^t \frac{1}{x + t_0} dx = \alpha(\ln(t + t_0) - \ln(k + 1 + t_0)).
$$

Thus, the product term becomes:

$$
\prod_{j=k+1}^{t-1} (1 - D_{2,1}\alpha_j) \leq \exp\left[ -D_{2,1}\alpha \left( \ln(t + t_0) - \ln(k + 1 + t_0) \right) \right] = \left( \frac{k + 1 + t_0}{t + t_0} \right)^{D_{2,1}\alpha}.
$$

Substituting back into the recursive inequality:

$$\mathbb{E}[L(q_t)] \leq \left(\frac{\bar{t}+t_0}{t+t_0}\right)^{D_{2,1}\alpha} \mathbb{E}[L(q_{\bar{t}})] + D_{2,2} \sum_{k=\bar{t}}^{t-1} \frac{\alpha^2}{(k+t_0)^2} \left(\frac{k+1+t_0}{t+t_0}\right)^{D_{2,1}\alpha}.$$

Now the summation term can be bounded by:

$$\sum_{k=\bar{t}}^{t-1} \frac{\alpha^2}{(k+t_0)^2} \left(\frac{k+1+t_0}{t+t_0}\right)^{D_{2,1}\alpha} \leq \sum_{k=t_0+\bar{t}}^{t+t_0-1} \frac{1}{k^2} \left(\frac{2k}{t+t_0}\right)^{D_{2,1}\alpha} = \frac{2^{D_{2,1}\alpha}}{(t+t_0)^{D_{2,1}\alpha}} \sum_{k=t_0+\bar{t}}^{t+t_0-1} k^{D_{2,1}\alpha-2}.$$

Define:

$$S(t) \doteq \sum_{k=t_0+\bar{t}}^{t+t_0-1} k^{D_{2,1}\alpha-2}.$$

The behavior of $S(t)$ depends on the value of $D_{2,1}\alpha$:

1. When $D_{2,1}\alpha < 1$, then $D_{2,1}\alpha - 2 < -1$. The sum $S(t)$ converges to a constant as $t \to \infty$:

$$S(t) \leq D_{2,3}.$$

   Therefore,

$$\frac{2^{D_{2,1}\alpha}S(t)}{(t+t_0)^{D_{2,1}\alpha}} \leq \frac{2^{D_{2,1}\alpha}D_{2,3}}{(t+t_0)^{D_{2,1}\alpha}}.$$

2. When $D_{2,1}\alpha = 1$, then $D_{2,1}\alpha - 2 = -1$. The sum $S(t)$ behaves like the harmonic series:

$$S(t) \leq \ln(t+t_0-1).$$

   Thus,

$$\frac{2^{D_{2,1}\alpha}S(t)}{t+t_0} \leq \frac{2^{D_{2,1}\alpha}\ln(t+t_0-1)}{t+t_0}.$$

3. When $D_{2,1}\alpha > 1$, then $D_{2,1}\alpha - 2 > -1$. The sum $S(t)$ grows polynomially:

$$S(t) \leq \frac{(t+t_0)^{D_{2,1}\alpha-1}}{D_{2,1}\alpha - 1}.$$

   Therefore,

$$\frac{2^{D_{2,1}\alpha}S(t)}{(t+t_0)^{D_{2,1}\alpha}} = \frac{2^{D_{2,1}\alpha}(t+t_0)^{D_{2,1}\alpha-1}}{(D_{2,1}\alpha-1)(t+t_0)^{D_{2,1}\alpha}} \leq \frac{2^{D_{2,1}\alpha}}{(D_{2,1}\alpha-1)(t+t_0)}.$$

Substituting the bounded summation term back into the recursive inequality, we obtain:

$$\mathbb{E}[L(q_t)] \leq \left(\frac{2t_0}{t+t_0}\right)^{D_{2,1}\alpha} \mathbb{E}[L(q_{\bar{t}})] + \frac{2^{D_{2,1}\alpha}S(t)}{(t+t_0)^{D_{2,1}\alpha}}$$
$$\leq \frac{D_{2,4}}{(t+t_0)^{D_{2,1}\alpha}}\mathbb{E}[L(q_{\bar{t}})] + \frac{D_{2,5}\ln(t+t_0)}{(t+t_0)^{\min(1,D_{2,1}\alpha)}}. \tag{22}$$

For the $\mathbb{E}[L(q_{\bar{t}})]$ term, since $\bar{t}$ is deterministic, following the similar derivation as above, we can obtain

$$\mathbb{E}[L(q_{\bar{t}})] \leq \left(\frac{t_0}{\bar{t}+t_0}\right)^{D_{2,1}\alpha} \mathbb{E}[L(q_0)] + D_{2,2} \sum_{k=0}^{\bar{t}-1} \frac{\alpha^2}{(k+t_0)^2} \left(\frac{k+1+t_0}{\bar{t}+t_0}\right)^{D_{2,1}\alpha}.$$

Similarly, the summation term can be bounded by:

$$\sum_{k=0}^{\bar{t}-1} \frac{\alpha^2}{(k+t_0)^2} \left( \frac{k+1+t_0}{\bar{t}+t_0} \right)^{D_{2,1}\alpha} \leq \frac{2^{D_{2,1}\alpha}}{(\bar{t}+t_0)^{D_{2,1}\alpha}} \sum_{k=t_0}^{\bar{t}+t_0-1} k^{D_{2,1}\alpha-2}$$

$$\leq \frac{2^{D_{2,1}\alpha}\bar{t}}{(\bar{t}+t_0)^{D_{2,1}\alpha}} \max(t_0^{D_{2,1}\alpha-2}, (\bar{t}+t_0-1)^{D_{2,1}\alpha-2}).$$

Therefore, $\mathbb{E}[L(q_{\bar{t}})] \leq D_{2,6}\mathbb{E}[L(q_0)] + D_{2,7}$. Combining this with (22), denote $B_{2,1} \doteq D_{2,4}D_{2,6}$, $B_{2,2} \doteq D_{2,1}$ and $B_{2,3} \doteq D_{2,4}D_{2,7} + D_{2,5}$ then completes the proof of the first case.

**Case 2:** $\epsilon_\alpha \in (0.5, 1)$. With the step size $\alpha_j = \frac{\alpha}{(j+t_0)^{\epsilon_\alpha}}$, the sum can be approximated by an integral:

$$\sum_{j=k+1}^{t-1} \alpha_j \geq \alpha \int_{k+1}^{t} \frac{1}{(x+t_0)^{\epsilon_\alpha}} dx = \frac{\alpha}{1-\epsilon_\alpha} \left[ (t+t_0)^{1-\epsilon_\alpha} - (k+1+t_0)^{1-\epsilon_\alpha} \right].$$

Thus, the product term becomes:

$$\prod_{j=k+1}^{t-1} (1-D_{2,1}\alpha_j) \leq \exp \left( -\frac{D_{2,1}\alpha}{1-\epsilon_\alpha} \left[ (t+t_0)^{1-\epsilon_\alpha} - (k+1+t_0)^{1-\epsilon_\alpha} \right] \right).$$

Substituting back into the recursive inequality, we can get:

$$\mathbb{E}[L(q_t)] \leq \exp \left( -\frac{D_{2,1}\alpha}{1-\epsilon_\alpha}(t+t_0)^{1-\epsilon_\alpha} \right) \exp \left( \frac{D_{2,1}\alpha}{1-\epsilon_\alpha}t_0^{1-\epsilon_\alpha} \right) \mathbb{E}[L(q_{\bar{t}})]$$

$$+ D_{2,2} \sum_{k=\bar{t}}^{t-1} \frac{\alpha^2}{(k+t_0)^{2\epsilon_\alpha}} \exp \left( -\frac{D_{2,1}\alpha}{1-\epsilon_\alpha} \left[ (t+t_0)^{1-\epsilon_\alpha} - (k+1+t_0)^{1-\epsilon_\alpha} \right] \right).$$

Denote $D_{2,6} \doteq \exp \left( \frac{D_{2,1}\alpha}{1-\epsilon_\alpha}t_0^{1-\epsilon_\alpha} \right)$. Now we bound the summation term. Since $2\epsilon'_\alpha > 1$, this term can be bounded by:

$$\sum_{k=\bar{t}}^{t-1} \frac{\alpha^2}{(k+t_0)^{2\epsilon'_\alpha}} \exp \left( -\frac{D_{2,1}\alpha}{1-\epsilon_\alpha} \left[ (t+t_0)^{1-\epsilon_\alpha} - (k+1+t_0)^{1-\epsilon_\alpha} \right] \right)$$

$$= \sum_{k=\bar{t}}^{\bar{t}+\lfloor\frac{t}{2}\rfloor-1} \frac{\alpha^2}{(k+t_0)^{2\epsilon'_\alpha}} \exp \left( -\frac{D_{2,1}\alpha}{1-\epsilon_\alpha} \left[ (t+t_0)^{1-\epsilon_\alpha} - (k+1+t_0)^{1-\epsilon_\alpha} \right] \right)$$

$$+ \sum_{k=\bar{t}+\lfloor\frac{t}{2}\rfloor}^{t-1} \frac{\alpha^2}{(k+t_0)^{2\epsilon'_\alpha}} \exp \left( -\frac{D_{2,1}\alpha}{1-\epsilon_\alpha} \left[ (t+t_0)^{1-\epsilon_\alpha} - (k+1+t_0)^{1-\epsilon_\alpha} \right] \right)$$

$$\leq \lfloor\frac{t}{2}\rfloor \frac{\alpha^2}{t_0^{2\epsilon'_\alpha}} \exp \left( -\frac{D_{2,1}\alpha}{1-\epsilon_\alpha} \left[ (t+t_0)^{1-\epsilon_\alpha} - (\bar{t}+\lfloor\frac{t}{2}\rfloor+t_0)^{1-\epsilon_\alpha} \right] \right) + \sum_{k=\bar{t}+\lfloor\frac{t}{2}\rfloor}^{t-1} \frac{\alpha^2}{(k+t_0)^{2\epsilon'_\alpha}}$$

$$\leq \lfloor\frac{t}{2}\rfloor \frac{\alpha^2}{t_0^{2\epsilon'_\alpha}} \exp(-D_{2,7}(t+t_0)^{1-\epsilon_\alpha}) + \int_{\bar{t}+\lfloor\frac{t}{2}\rfloor-1}^{t-1} \frac{\alpha^2}{(x+t_0)^{2\epsilon'_\alpha}} dx$$

$$\leq \lfloor\frac{t}{2}\rfloor \frac{\alpha^2}{t_0^{2\epsilon'_\alpha}} \exp(-D_{2,7}(t+t_0)^{1-\epsilon_\alpha}) + \frac{\alpha^2}{1-2\epsilon'_\alpha}(t+t_0)^{1-2\epsilon'_\alpha}$$

$$\leq D_{2,8}(t+t_0)^{1-2\epsilon'_\alpha}.$$

Substituting the bounded summation term back into the recursive inequality, we obtain:

$$\mathbb{E}[L(q_t)] \leq \exp \left( -\frac{D_{2,1}\alpha}{1-\epsilon_\alpha}(t+t_0)^{1-\epsilon_\alpha} \right) D_{2,6}\mathbb{E}[L(q_{\bar{t}})] + D_{2,2}D_{2,8}(t+t_0)^{1-2\epsilon'_\alpha}$$

$$\le \exp\left(-\frac{D_{2,1}\alpha}{1-\epsilon_\alpha}(t+t_0)^{1-\epsilon_\alpha}\right)D_{2,6}\mathbb{E}[L(q_{\bar{t}})] + D_{2,9}(t+t_0)^{1-2\epsilon'_\alpha}.$$

Then Theorem 2 follows from using the Gronwall inequality to bound $L(q_{\bar{t}})$ with $\|q_0\|$ and the equivalence of norms, with $B_{2,4}$, $B_{2,5}$, and $B_{2,6}$ selected similarly. □

## E. Comparsion with other algorithms

We compare our unmodified linear $Q$-learning algorithm with several variants that incorporate common modifications. All experiments are conducted in the Baird's counterexample environment with 10 independent runs per algorithm. All algorithms use the same $\epsilon$-softmax behavior policy with adaptive temperature parameter $\kappa_0 = 10$ and $\epsilon = 0.1$ as described in our main paper.

We compare the following algorithms:

1. **No Modification:** The original linear $Q$-learning algorithm as analyzed in our theoretical results.

2. **Target Network**: A separate target network is used for computing the TD error, with the target network updated every 10 timesteps.

3. **Weight Projection:** After each update, the weights are projected onto a ball with radius 10, ensuring $\|w_t\|_2 \le 10$ at all times(Chen et al., 2023).

4. **Ridge Regularization:** A ridge regularization term is added to the update rule with coefficient $\eta = 0.01$(Zhang et al., 2021), penalizing large weight values.

Figure 2 shows the evolution of weight norms $\|w_t\|_2^2$ for all four algorithms. While all methods eventually maintain bounded weights, our unmodified approach achieves comparable performance without the computational overhead or hyperparameter tuning required by the other methods.

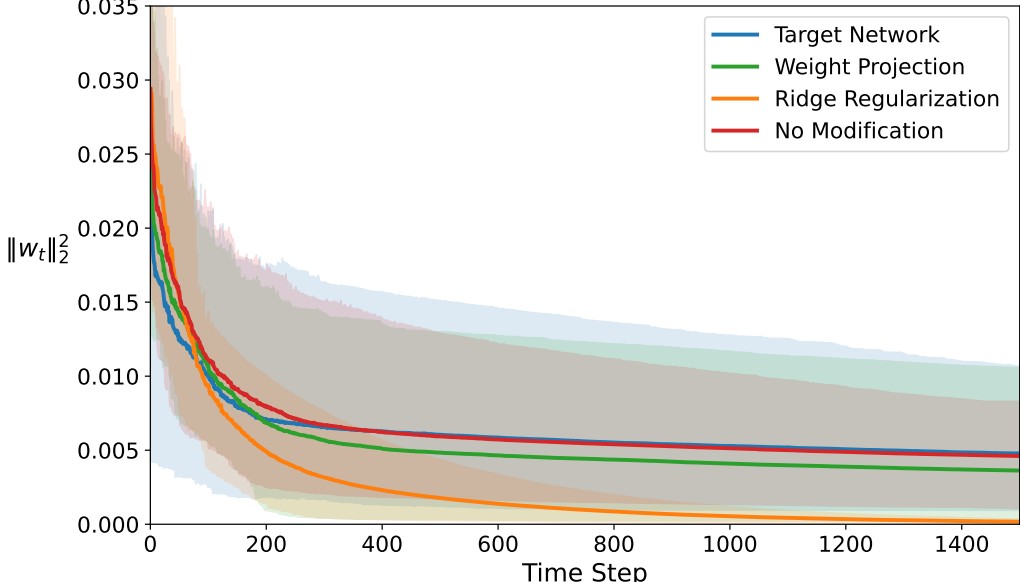

*Figure 2.* Each scenario is independently run 10 times, with the solid lines representing the averages of the squared $L^2$ norm of weights, and the shaded areas indicating the ranges between minimum and maximum values. This comparison illustrates the impact of different modification strategies on the algorithm's convergence behavior.

