# OpenReview forum: "Linear $Q$-Learning Does Not Diverge in $L^2$: Convergence Rates to a Bounded Set"
_ICML.cc/2025/Conference — ICML 2025 poster_

### Official Review · Reviewer_DQiq · 2025-03-10

**Overall Recommendation:** 4

**Summary:**

This paper challenges the widely held belief that linear Q-learning can diverge and instead proves that linear Q-learning converges to a bounded set. Unlike previous studies that required algorithmic modifications (e.g., target networks, experience replay), this work establishes the first L2 convergence rate for linear Q-learning without modifying the original algorithm. The key technique is a novel stochastic approximation result under fast-changing Markovian noise, proving that linear Q-learning remains stable when using an ϵ-softmax behavior policy with an adaptive temperature. The paper also extends the results to tabular Q-learning, establishing its L2 convergence rate for the first time.

**Claims And Evidence:**

The paper claims that:
1.	Linear Q-learning does not diverge, but instead converges to a bounded set.
2.	An L2 convergence rate for linear Q-learning is derived without algorithmic modifications.
3.	Tabular Q-learning also has an L2 convergence rate, confirmed via a novel pseudo-contraction property.
The results are well-supported theoretically.

**Essential References Not Discussed:**

The references in the paper are appropriate.

**Experimental Designs Or Analyses:**

The paper does not include empirical experiments but instead evaluates the method using:
1.	Theoretical analysis: Proving L2 convergence rates mathematically.
2.	Comparisons with prior work, highlighting weaker assumptions and stronger guarantees.
3.	Algorithmic stability proofs without relying on experience replay, target networks, or weight projection.

**Methods And Evaluation Criteria:**

The paper introduces:
1.	A novel stochastic approximation framework to analyze single-timescale Markovian noise.
2.	ϵ-softmax behavior policy with adaptive temperature to ensure sufficient exploration.
3.	L2 convergence analysis for both linear and tabular Q-learning.
4.	A pseudo-contraction property of the weighted Bellman operator, proving tabular Q-learning’s stability.
Evaluation is based on:
•	Theoretical convergence proofs.
•	Comparisons with prior work, showing that previous analyses required restrictive assumptions.
•	Mathematical guarantees on stability and error bounds.

**Other Comments Or Suggestions:**

•  It would be helpful to update the references to reflect their most recent status. In other words, some cited papers have since been published in journals or presented at conferences, and their reference details should be updated accordingly.
•  Improving the readability of the presentation would be beneficial, especially given the extensive mathematical content.
•  It would be valuable to include experiments to demonstrate the results. In particular, testing the well-known Baird example of the deadly triad would be interesting.
•  It would be helpful to provide an intuitive explanation for the choice of the specific exploration policy using softmax.

**Other Strengths And Weaknesses:**

Strengths:
•	First L2 convergence rate for linear Q-learning.
•	Strong theoretical results without modifying the algorithm.
•	Novel single-timescale analysis of Markovian noise.
Weaknesses:
•	No empirical validation

**Questions For Authors:**

Comparing the convergence rate for the tabular case with existing rates in the literature would be useful. Is the proposed method faster, or is its performance simply comparable to existing approaches?

**Relation To Broader Scientific Literature:**

This work contributes to:
•	Q-learning stability analysis: It disproves prior beliefs that linear Q-learning can diverge.
•	Stochastic approximation theory: Extends single-timescale convergence analysis.
•	Reinforcement learning theory: Strengthens understanding of off-policy learning stability.

**Theoretical Claims:**

The paper establishes:
1.	Linear Q-learning is stable: It does not diverge but converges to a bounded set.
2.	L2 convergence rates: Error bounds decrease over time, ensuring stability.
3.	Tabular Q-learning also has an L2 convergence rate, supported by a novel pseudo-contraction property.
4.	Single-timescale stochastic approximation: Unlike prior two-timescale methods, the analysis holds even when transition functions evolve at the same rate as weights.
The mathematical proofs are rigorous.

---

> ### Author Rebuttal · Authors · 2025-03-31
>
> Thank you for your positive evaluation and perfect score for our manuscript. We appreciate the opportunity to clarify the comparison of our tabular $Q$-learning convergence rate with existing literature:
> > Comparing the convergence rate for the tabular case with existing rates in the literature would be useful. Is the proposed method faster, or is its performance simply comparable to existing approaches?
>
> Comparing our tabular $Q$-learning convergence rate (Theorem 2) with existing literature reveals that our method achieves an $L^2$ rate simply comparable to prior approaches. Theorem 2 shows exponential decay, similar to rates in works like Even-Dar et al. (2003) and Chen et al. (2021), which also exhibit exponential convergence under different assumptions (e.g., count-based rates or fixed policies).
>
> The primary novelty of our result lies not necessarily in achieving a strictly faster rate in terms of the exponent's constants (as comparing these constants, which depend on complex problem parameters like $B_{2,5}$ and $B_{2,6}$  across different analytical frameworks is inherently difficult), but rather in demonstrating this competitive exponential convergence under a practical adaptive $\epsilon$-softmax behavior policy and without using count-based learning rates.
>
> Therefore, while the exponential convergence form aligns with existing benchmarks, the significance stems from achieving this strong type of guarantee under arguably more standard algorithmic settings used in practice. We view our rate as highly competitive, prioritizing the demonstration of robust convergence under these practical conditions.
>
> > It would be helpful to update the references to reflect their most recent status.
>
> Thank you for the thoughtful and valuable suggestions. We appreciate the recommendation to update references to their latest status and will ensure all citations reflect the most recent publication details.
>
> > Improving the readability of the presentation would be beneficial, especially given the extensive mathematical content.
>
> Improving readability is also a great point, and we will refine the presentation to make the mathematical content more accessible. For example, we will add detailed derivations in the revised Appendix (e.g., Sec C.4) showing how intermediate constants like $D$ combine, considering factors like norm conversions, to form the final $B$ bounds presented in the main theorems.
>
> > It would be valuable to include experiments to demonstrate the results. In particular, testing the well-known Baird example of the deadly triad would be interesting.
>
> We do agree experiments would be a great add-on. However, we note that we study exactly the same algorithm as Meyn (2024) and Meyn (2024) already includes extensive experimental validation, including Baird example. We feel there is no need to redo it, as our work focuses on novel convergence rates that complement Meyn's empirical results.
>
> > It would be helpful to provide an intuitive explanation for the choice of the specific exploration policy using softmax.
>
> We’re grateful for the suggestion to explain the softmax exploration policy intuitively. The adaptive $\epsilon$-softmax policy choice is technically motivated. Our analysis framework (Theorem 3) requires Lipschitz continuity of the expected update $h(w)$ (Assumption A3/A3'). Our policy ensures this (Lemmas 5, 6), enabling the analysis. Standard $\epsilon$-greedy is discontinuous due to `argmax` and typically violates this assumption, preventing the direct application of our proof technique. The adaptive temperature $\kappa_w$ is the crucial mechanism ensuring this smoothness holds globally, even if $\\|w\\|$ is large (by making the policy appropriately less sensitive). This is also key to our proof technique. An $\epsilon$-greedy policy would fail to use our novel single-timescale SA result. We will clarify this more in the revision.
>
> We truly appreciate your support and positive evaluation of our research.

---

### Official Review · Reviewer_2Mh8 · 2025-03-11

**Overall Recommendation:** 3

**Summary:**

This paper provides the first L² convergence rate analysis for linear Q-learning with no algorithmic modifications. The authors show that linear Q-learning converges to a bounded set without requiring target networks, weight projection, experience replay, or regularization techniques that are typically employed to ensure stability. The key innovation is using an ϵ-softmax behavior policy with an adaptive temperature parameter. The paper also establishes convergence rates for tabular Q-learning with an ϵ-softmax behavior policy. The technical approach leverages a novel general result on stochastic approximations under Markovian noise with fast-changing transition functions.

**Claims And Evidence:**

The paper makes two primary claims:

1. Linear Q-learning with an ϵ-softmax behavior policy and adaptive temperature converges to a bounded set with a provable L² convergence rate.
2. Tabular Q-learning with an ϵ-softmax behavior policy has a provable L² convergence rate to the optimal action-value function.

These claims are rigorously supported by theoretical analysis and proofs. The first claim extends recent work by Meyn (2024), which established almost sure convergence to a bounded set but did not provide a convergence rate. The second claim appears to be novel in establishing convergence rates for tabular Q-learning with an ϵ-softmax policy and without count-based learning rates.
The evidence is primarily theoretical, with detailed mathematical proofs for both claims. The authors also provide comparative tables that position their work against existing literature, highlighting how their analysis avoids algorithm modifications and restrictive assumptions that previous works relied upon.

**Essential References Not Discussed:**

No

**Experimental Designs Or Analyses:**

As mentioned, the paper does not include empirical evaluation, which is a notable weakness.

**Methods And Evaluation Criteria:**

The paper's methodology is primarily theoretical and builds on established techniques for analyzing stochastic approximation algorithms. Key methodological contributions include:

1. A general stochastic approximation result for time-inhomogeneous Markovian noise in a single-timescale setting, which is more challenging than the two-timescale settings analyzed in prior work.
2. The identification of a pseudo-contraction property for the weighted Bellman optimality operator, which enables the convergence analysis for tabular Q-learning.
3. Technical innovations in bounding terms involving adaptive temperature in the ϵ-softmax policy.

The paper does not include empirical evaluation, which is a limitation. While the theoretical contributions are significant, experimental validation would strengthen the work by demonstrating practical convergence behavior and comparing it to existing approaches with algorithmic modifications.

**Other Comments Or Suggestions:**

No.

**Other Strengths And Weaknesses:**

Additional strengths:

- The paper's analysis is technically sophisticated yet clearly presented.
- The focus on the original Q-learning algorithm without modifications is valuable for understanding fundamental convergence properties.
- The general stochastic approximation result (Theorem 3) may have applications beyond Q-learning.

Additional weaknesses:

- The paper does not discuss the implications of converging to a bounded set rather than to the optimal policy or value function for linear Q-learning.
- There is limited discussion of the practical significance of the ϵ-softmax behavior policy with adaptive temperature compared to more commonly used ϵ-greedy policies.
- The paper could benefit from more intuitive explanations of why the adaptive temperature mechanism is crucial for convergence.

**Questions For Authors:**

1. Could you provide empirical evidence to validate the theoretical convergence rates and compare the performance with methods that use algorithmic modifications?
2. How should practitioners select the hyperparameters ϵ and κ0 to ensure convergence in practical applications?
3. For linear Q-learning, what can be said about the quality of the policies derived from the bounded set to which the algorithm converges? How far might these policies be from optimal?
4. How does the proposed approach scale to high-dimensional state spaces or function approximation methods beyond linear?
5. Could the analysis be extended to other off-policy algorithms such as Expected SARSA or Q(λ)?
6. How tight are the derived bounds, and do you believe the convergence rates are optimal?


### Post-rebuttal response ###
I appreciate the authors' thorough responses to my questions. While they've addressed my technical concerns regarding comparisons to Meyn (2024), hyperparameter selection, and theoretical implications, I maintain my original score. My assessment balances the paper's strong theoretical contribution against the absence of empirical validation and limited insight into policy quality. Though the authors committed to adding algorithmic comparisons, these aren't yet in the manuscript. The paper deserves publication, but these limitations warrant my original assessment.

**Relation To Broader Scientific Literature:**

Tables 1 and 2 provide comprehensive comparisons with previous work on linear and tabular Q-learning, clearly highlighting the novel aspects of this research.

**Theoretical Claims:**

The paper's theoretical claims are generally well-founded and rigorously proven. The proofs follow a logical structure, first establishing a general stochastic approximation result (Theorem 3) that is then applied to both linear Q-learning (Theorem 1) and tabular Q-learning (Theorem 2).
Strengths:

- The analysis handles the challenging single-timescale case where the transition matrix evolves as fast as the weights.
- The paper identifies the novel pseudo-contraction property of the weighted Bellman optimality operator.
- The convergence rates are explicit and account for different learning rate regimes.

Limitations:

- The constants in the convergence bounds (B1,3, B1,6, B2,3, B2,6) are not fully characterized, making it difficult to assess the tightness of the bounds.
- The analysis requires sufficiently small ϵ and sufficiently large κ0 and t0, but does not provide explicit guidance on how to set these hyperparameters in practice.
- The paper lacks discussion on the optimality of the obtained convergence rates.

---

> ### Author Rebuttal · Authors · 2025-03-31
>
> We're grateful for your thoughtful inquiry and positive assessment. We address your points below:
>
> > Could empirical evidence be provided ...?
>
> This paper studies the same algorithm as Meyn (2024), which already provides extensive empirical results on its behavior. So we feel there is no need to redo it again.
>
> > ... compare performance with methods using algorithmic modifications.
>
> Thanks for the great suggestion. We will add this comparison in the next version and believe this will be a great add-on. While our theoretical contributions are already significant, this addition should certainly enhance the overall paper.
>
> > How should practitioners select $\epsilon$, $\kappa_0$, and $t_0$?
>
> We appreciate this practical query. Since our algorithm aligns with Meyn (2024), practitioners can draw from its empirical insights: $\epsilon \leq 0.2$ ensures stability for high $\gamma$ (e.g., 0.99), per Lemma A.9; $\kappa_0=1$ serves as a reasonable starting point; and for our step size $\alpha_t=\alpha/(t+t_0)^{\epsilon_\alpha}$, we suggest $t_0 \geq 10$ with $\epsilon_\alpha=0.85$ for balance. Our revision will clarify these, blending Meyn’s findings with our theoretical framework.
>
> > For linear $Q$-learning, what are the implications of converging to a bounded set and the quality of policies?
>
> Thanks for this insightful question. The immediate implication is that we now know that the variance of the parameters is uniformly bounded across time steps. Assessing the quality of the learned policy is challenging without making further artificial assumptions (see Meyn (2024) for more discussion) and is indeed an opportunity for future work.
>
> > How does it scale to high dimensions or non-linear methods?
>
> The approach scales to high-dimensional linear settings. Theorem 1 holds regardless of dimension $d$. The adaptive temperature $\kappa_w$ is crucial, ensuring sufficient smoothness (Lemmas 5, 6) and controlled exploration even if weight norm $\\|w\\|_2$ grows large.
>
> Extending to non-linear methods (e.g., neural networks) is feasible if we would like to introduce overparameterized networks and do linearization around initial weights (cf. Cai (2019)). We will additionally have some terms controlled by the width of the network. We believe such an extension is mostly a matter of labor and won't shed new technical insights.
>
> Ref: Q. Cai et al. Neural Temporal-Difference Learning Converges to Global Optima. Advances in Neural Information Processing Systems, 32, 2019.
>
>
> > Could it extend to Expected SARSA or Q($\lambda$)?
>
> For Expected SARSA, Yes. Replacing $\max_a$ with an expectation over $\mu_w$ (for on-policy) or some $\pi_w$ (for off-policy) fits our framework (Theorem 3), maintaining Lipschitz continuity (Lemma 5).
> For Q($\lambda$), the existence of off-policy eligibility traces will significantly complicate the behavior of the chain (we have to construct an auxiliary chain to contain the trace). But we envision the techniques from Yu (2012) can help.
>
> Ref: H. Yu. Least squares temporal diﬀerence methods: An analysis under general conditions. SIAM Journal on Control and Optimization, 2012.
>
> > The constants in the bounds are not fully characterized. How tight/optimal are the rates?
>
> Thank you for raising this point. We will revise the Appendix to show how intermediate constants(e.g., $D$) combine into the final $B$ bounds presented in the main theorems, accounting for norm conversions.
>
> Tightness is unclear due to unknown parameters and RL’s hardness with linear function approximation (cf. Liu (2023)). We offer the first $L^2$ rates for linear Q-learning under weak assumptions as a benchmark. Liu (2023) suggests polynomial convergence is unlikely, supporting our state-of-the-art rates. For tabular Q-learning, we improve prior work with an $\epsilon$-softmax policy, without count-based steps. Optimality remains open.
>
> Ref: Liu, S. et al. Exponential hardness of reinforcement learning with linear function approximation. In The Thirty Sixth Annual Conference on Learning Theory (pp. 1588-1617). PMLR.
>
> > Limited discussion on adaptive $\epsilon$-softmax vs $\epsilon$-greedy and why adaptive temperature is crucial.
>
> The motivation for not using the $\epsilon$-greedy policy is detailed at the very end of our response to the 2nd reviewer (bV3r). In short, it's because `argmax` is discontinuous.
>
> The adaptive temperature $\kappa_w$ further ensures the required smoothness beyond continuity, even when $\\|w\\|$ is large. This smoothness allows us to apply our analysis framework (Theorem 3) to prove $L^2$ rates. We will add this discussion in the revision.
>
> Thank you again for the constructive feedback and positive evaluation. We hope these responses have adequately addressed your insightful questions.

---

### Official Review · Reviewer_bV3r · 2025-03-15

**Overall Recommendation:** 4

**Summary:**

The paper provides a theoretical analysis of Q-learning with linear approximation and a tabular setting.

**Claims And Evidence:**

The claims made in the manuscript are clear and well-written.

**Essential References Not Discussed:**

Relevant and important literatures are nicely summarized and provided.

**Experimental Designs Or Analyses:**

No numerical experiments.

**Methods And Evaluation Criteria:**

No numerical experiments.

**Other Comments Or Suggestions:**

See the questions.

**Other Strengths And Weaknesses:**

Strength: The paper is well-structured and written very clearly for readers to follow. The theoretical result they establish would gain interest among the RL community.

Weakness: There are still some remaining things to be polished and details to fill in. Please see the questions.

**Questions For Authors:**

I have the following set of questions, which would help further determine my opinion on this manuscript. I really liked the manuscript, and the current rating won't be my final rating as long as the authors address the questions below.

1. I was at first confused by the statement of Assumption A1, as this sounded more like a non-trivial lemma. I later realized this is Lemma 9 of Zhang from the appendix. However, when I check Zhang (2021) Lemma 9, it doesn't seem like the citation is correct. Can you please verify this?

2. In the proof of  Lemma 6, the authors use the contraction property of the Bellman optimality operator $\mathcal{T}$. If I am not mistaken, this operator is contraction with respect to $\ell_\infty$-norm, and authors are invoking the equivalence of the norm in finite-dimensional Euclidean space. So for the argument in Lemma 6 to go through, it seems that the finiteness of state space $S$ and action space $A$ is must; is this correct? Also, could you please

3. I couldn't quite follow the derivation of the ultimate bound in the proof of 7. It seems that the bound $C_7 \|w\|_2$, which works for $\|w\|_2 \le 1$ case, is larger than the penultimate bound. Can you please clarify this step?

4. On page 19, in the first term of the RHS in expression between line 1023-1025, shouldn't it be $(\frac{t_0 + \bar t}{t+t_0})^{D_{1,2}\alpha}$ instead of $(\frac{t_0}{t+t_0})^{D_{1,2}\alpha}$ ?

5. Can you also please provide the details on how the condition of Gronwall's inequality is met for the last step of the Theorem 1 proof? I wonder how one is able to make sure the condition holds for all $n$ in Grownwall's inequality.

6. On page 20, in lines 1087-1092, I believe this inequality doesn't hold as $1//k^{\epsilon} > 1//(k+1)^{\epsilon}$. It should be an easy fix though, by introducing some constant factor.

7. Could you also please highlight which part of the proof would go wrong with the epsilon-greedy policy? I think this would also strengthen the paper.

**Relation To Broader Scientific Literature:**

The contribution of this paper is nontrivial since it resolves a long-standing problem of convergence of Q-learning in a more realistic setting, i.e., using a soft-max policy without assuming a strong assumption on the policies, i.e., Lipschitz policy.

**Theoretical Claims:**

Please see questions.

---

> ### Author Rebuttal · Authors · 2025-03-31
>
> Thank you for the constructive feedback and careful reading of our manuscript, which has improved our manuscript. We will incorporate your suggestions into the revised version. Below are our responses:
>
> > Assumption A1 seemed non-trivial, and the citation to Zhang (2021) Lemma 9 appears incorrect.
>
> We apologize for the confusion. In short, Lemma 9 of Zhang (2021) is not meant to verify A1.
>
> A1 is much more trivial than it looks like. To our knowledge, A1 has been used at least three times previously: (1) A3.1 and A3.2 in Zhang et al. (2022) JMLR, (2) A5.1 in Zhang et al. (2021) ICML, and (3) Marbach & Tsitsiklis (2001) (TAC) (same concept with different wording). A1 is readily satisfied under simple conditions, e.g., when our Assumption A3.1 (ergodicity under the uniform random policy) holds and an exploring policy (like $\epsilon$-greedy or $\epsilon$-softmax with $\epsilon > 0$) is used. The intuition is that $\epsilon > 0$ ensures that all the state-action transition matrices in the closure share the same connectivity, i.e., any policy in that closure will choose all actions with a strictly positive probability. So all transition matrices share the same ergodicity with the uniform random policy. Our verification of A1 is in Line 332 - 346. This is also how Zhang et al. (2021, 2022) verify this assumption, see more discussion in Zhang (2022) in the text below their A3.2 and A4.4. We will add more details in the revision.
>
> Lemma 9 of Zhang (2021) states the Lipschitz continuity of the stationary distribution w.r.t. the policy parameters under their A5.1. We restate it as our Lemma 15. We will clarify this further in our revision.
>
> Ref: Marbach, P. and Tsitsiklis, J. N. Simulation-based optimization of markov reward processes. IEEE Transactions on Automatic Control, 2001.
>
> > Lemma 6 uses norm equivalence; does this require finite state and action spaces, and is this assumption necessary?
>
> Yes and yes. Analyzing the underlying time-inhomogeneous Markov chain in a general state space is inherently difficult, even if it is compact. We need to introduce lots of (hard to verify) assumptions to ensure uniform mixing and will complicate the presentation a lot, especially as key constants like $β$ depend on $\log |\mathcal{A}|$, posing challenges for infinite action spaces.
>
> > The final bound in Lemma 7's proof seems inconsistent, particularly for the case where $\\|w\\|\_2 < 1$.
>
> Thanks for spotting this oversight. In the case when $\\|w\\|\_2 \leq 1$, we have $\\|w\\|^2\_2\leq \\|w\\|\_2$, which then gives us $\langle w, A(w)w + b(w)\rangle \leq (C_{7,1} + C_{7,2})\\|w\\|\_2 \leq -\beta \\|w\\|^2\_2 + (C_{7,1} + C_{7,2} + \beta)\\|w\\|\_2$. Thus, we can redefine $C_7 = C_{7,1} + C_{7,2} + \beta$ to ensure
> $\langle w, A(w)w + b(w)\rangle \leq -\beta \\|w\\|^2\_2 + C_7\\|w\\|\_2$ for both cases. We will correct this definition in the next version.
>
>
> > Typo in an exponent's base on page 19.
>
> You are correct. The base of the exponent should be $(\frac{t_0+\bar{t}}{t+t_0}) ^{D_{1,2}\alpha}$. We will correct the term and subsequent constant $D_{1,3}$ in the revised manuscript.
>
>
> > Details are needed on how the conditions for Gronwall's inequality are met in the final step of Theorem 1's proof.
>
> Thank you for questioning about the proof details. Below, we provide a detailed explanation to address this concern, and we will include this in Section C.4 of the revised manuscript.
> Starting from the update of $w_{t+1}$, we have $\\|w_{t+1}\\|\leq \\|w_t\\| + \alpha_t \\|H(w_t,Y_{t+1})\\|\\\\
> \leq \\|w_t\\| + \alpha_t C_{18}(\\|w_t\\|+1)$.
>
> That is, $\\|w_{t+1}\\| \leq \alpha_0 C_{18} + \sum_{i=0}^t(\alpha_0 C_{18}+1)\\|w_i\\|$. Applying discrete Gronwall inequality, we obtain
> $\\|w_{\bar{t}}\\| \leq (C_{18} + \\|w_0\\|) \exp(\sum_{t=0}^{\bar{t}-1} (1+\alpha_0 C_{18})) = (C_{18} + \\|w_0\\|) \exp(\bar{t}+\bar{t}\alpha_0 C_{18})$.
>
> Furthermore, combining this with the bound on $B_{1,3}$ from Section C.4, we have $B_{1,3} = 2\left(\frac{D_{1,3}}{(t + t_0)^{D_{1,2} \alpha}}\times 2C_{18} \exp(2\bar{t}+2\bar{t}\alpha_0  C_{18})+D_{1,4}\right)$, where $D_{1,4}$ is a constant bounded in Section C.4. We will include this detailed derivation in the revised manuscript.
>
>
> > An inequality involving $\frac{1}{k^\epsilon}$ on page 20 seems incorrect.
>
> Thank you for this precise catch. We will correct it by introducing an appropriate constant factor $2^{\epsilon_\alpha}$ and update Appendix C.4 accordingly.
>
> > It would strengthen the paper to explain why the analysis doesn't apply to epsilon-greedy policies.
>
> We agree with this suggestion. The `argmax` makes the $\epsilon$-greedy policy discontinuous. This means the stationary distribution is also discontinuous in $w$, violating the Lipschitz continuity needed for $h(w)$ (Assumption A3/A3'). Our adaptive $\epsilon$-softmax policy is smooth by design, satisfying this requirement.
>
> We hope these responses address your points and you could consider updating your evaluation.

---

### Official Review · Reviewer_vZu3 · 2025-03-18

**Overall Recommendation:** 4

**Summary:**

The authors propose the analysis of Q-learning with linear functional approximation that identifies under an assumption of $\varepsilon$-softmax parametrization with an adaptive temperature that keeps the norm of the logits fixed, the rate of convergence to a bounded region.  Additionally, the authors provided the analysis for a case of finite MDPs and showed the algorithm's convergence to the optimal Q-value.

**Claims And Evidence:**

1. The authors established the first $L^2$ convergence rate of linear Q-learning to a bounded set.

This result has been proven in Theorem 1. However, the dependence of the target weight norm on a number of state-action pairs is enormous, and it is especially harmful in the case of linear functional approximation since it is used in the case of an infinite number of state-action pairs. Additionally, the value of $\beta$ (it is important since convergence rates depend on $\exp(\beta)$) was never specified in the paper with dependence on $\kappa$ and $\varepsilon$.

2. The authors provide an analysis for a general stochastic approximation with time-inhomogeneous Markov noise under a novel type of assumption.

The authors provided the proof of the result in Theorem 3. While Assumption 3 might be seen as a strong one, the authors showed that it holds in the case of specific adaptive softmax parameterization in Linear Q-learning.

**Essential References Not Discussed:**

I think that for a general exposition of the problem, it is important to discuss the hardness results of the proposed setting, i.e., (Lui et al., 2023). In particular, the fact of NP-hardness of finding the optimal policy under linear Q*-assumption gives a hint that convergence to the optimal policy in polynomial time should not be possible.


Liu, S., Mahajan, G., Kane, D., Lovett, S., Weisz, G., & Szepesvári, C. (2023, July). Exponential hardness of reinforcement learning with linear function approximation. In The Thirty Sixth Annual Conference on Learning Theory (pp. 1588-1617). PMLR.

**Experimental Designs Or Analyses:**

N/A

**Methods And Evaluation Criteria:**

N/A

**Other Comments Or Suggestions:**

There is a discrepancy in notation between the main text and Appendix: in Appendix, the constants are denoted by a letter $D$, whereas in the main text, the letter $B$ is used.

**Other Strengths And Weaknesses:**

Given the large value of the constant that bounds the maximal possible value, this result does not provide a lot of insights into the quality of the final solution. For me, these convergence rates do not give any additional insights into the algorithm behavior beyond the asymptotical result of Meyn (2024).

**Questions For Authors:**

- Is it possible to provide some bounds on the value of a constant $\beta$?
- Could you provide an exact value of the constant $B_{1,3}$ in the main text?

**Relation To Broader Scientific Literature:**

The paper can find its place in the current literature on stochastic approximation of Q-learning with linear functional approximation.

**Theoretical Claims:**

I did not check the theoretical results in detail. The main idea of the main result (Theorem 3) (stepping back on mixing time and coupling two chains, one the original and the second one is homogeneous) looks reasonable to me and should give the desired result, given Assumption 3 or 3'.

---

> ### Author Rebuttal · Authors · 2025-03-31
>
> Thank you for your valuable feedback and positive evaluation. We appreciate the opportunity to address the specific questions and comments.
> > Hardness results in Liu (et al., 2023).
>
> Thanks for the excellent suggestion. The hardness results indeed provide key context and may suggest that convergence to a set with the current rate might be the best we can hope. We will add this discussion and cite Liu et al. (2023) in Section 4 in next version.
>
> >  Large constant ... making the non-asymptotic rates potentially no more informative than Meyn's (2024) asymptotic result.
>
> We understand the concern but want to make two points. (1) Meyn's results are only almost sure boundedness and convey no information about the variance. Even ignoring the rates, our results still provide a uniform bound of the variance across time steps. We argue that this is already valuable. (2) Although the constant is large, we argue it's the first result of this kind. It provides an explicit exponential decay rate, showing the dependency on key factors like $\gamma$, step-size, and exploration parameters ($\epsilon$, $\kappa_0$). We believe this will lay a foundation for future refinements.
>
> >  Notation discrepancy for constants between the main text (using B) and the Appendix (using D).
>
> Thanks for noting this. We use $B$ for main theorem constants and $D$ for intermediate ones in the Appendix. Some translation uses the equivalence between norms. We will clarify all these in next version.
>
> > Is it possible to provide some bounds on the value of a constant $\beta$?
>
> We appreciate the suggestion. We adopt the explicit bound for $\beta$ from Meyn (2024), Lemma A.9: $\beta = \left[ (1 - \gamma) - \epsilon \gamma \sqrt{\epsilon^{-1} + (1 - \epsilon)^{-1}} \right] \lambda_{\min}(X^\top D_{\mu_w} X) - \gamma (1 - \epsilon) \frac{\log (|\mathcal{A}|)}{\kappa_0} \sqrt{\lambda_{\max}(X^\top D_{\mu_w} X)}$, where $\lambda_{\min}(\cdot)$ and $\lambda_{\max}(\cdot)$ namely denote the minimal and maximal eigenvalue of the matrix. This bound is positive for sufficiently small $\epsilon$ and large $\kappa_0$. We will add this to the revised manuscript (Appendix C.3).
>
> > Could you provide an exact value of the constant $B_{1,3}$ in the main text?
>
> Thank you for the suggestion. We have $B_{1,3} \leq 2\left(\frac{D_{1,3}}{(t + t_0)^{D_{1,2} \alpha}}\times 2C_{18} \exp(2\bar{t}+2\bar{t}\alpha_0  C_{18})+D_{1,4}\right)$, where the parameters $D_{1,3}$, $D_{1,2}$, $C_{18}$, $D_{1,4}$, $t_0$, and $\alpha$ are fixed constants from our analysis. Those constants are well defined and we can easily assemble them. We will add this in the revision but unfortunately we cannot display it here right now because it's too long.
>
> Thanks again for your helpful questions, which have allowed us to enhance the technical clarity of our manuscript. We hope these responses fully address your points.

---

> > ### Comment · Reviewer_vZu3 · 2025-04-02
> >
> > I would like to thank the authors for their response and for writing down the expressions of the constants' values. My main concern was addressed, and I will happily increase my score.

---

### Decision · Program_Chairs · 2025-05-01

**Decision:**

Accept (poster)

**Comment:**

The authors did a good job of addressing the concerns raised by the reviewers, and everybody converged to a positive recommendation on this one. I encourage the authors to clarify any issues that caused confusion among the reviewers in their final version.